# Online Locally Differentially Private Conformal Prediction via Binary Inquiries

**Qiangqiang Zhang**[1][*]   **Chenfei Gu**[2][*]   **Xinwei Feng**[1]    **Jinhan Xie**[3][†]   **Ting Li**[2][†]

[1] Zhongtai Securities Institute for Financial Studies, Shandong University
[2] School of Statistics and Data Science, Shanghai University of Finance and Economics
[3] Yunnan Key Laboratory of Statistical Modeling and Data Analysis, Yunnan University

## Abstract

We propose an online conformal prediction framework under local differential privacy to address the emerging challenge of privacy-preserving uncertainty quantification in streaming data environments. Our method constructs dynamic, model-free prediction sets based on randomized binary inquiries, ensuring rigorous privacy protection without requiring access to raw data. Importantly, the proposed algorithm can be conducted in a one-pass online manner, leading to high computational efficiency and minimal storage requirements with $\mathcal{O}(1)$ space complexity, making it particularly suitable for real-time applications. The proposed framework is also broadly applicable to both regression and classification tasks, adapting flexibly to diverse predictive settings. We establish theoretical guarantees for long-run coverage at a target confidence level, ensuring statistical reliability under strict privacy constraints. Extensive empirical evaluations on both simulated and real-world datasets demonstrate that the proposed method delivers accurate, stable, and privacy-preserving predictions across a range of dynamic environments.

## 1   Introduction

Modern decision-making systems are increasingly deployed in streaming environments such as real-time bidding, health monitoring, and algorithmic trading, where data distributions evolve over time. In these dynamic settings, reliable uncertainty quantification is essential for ensuring trustworthy predictions and maintaining robustness under distributional shift. Conformal prediction (CP) [Vovk et al., 2005] provides a powerful, distribution-free framework for constructing prediction sets with formal finite-sample coverage guarantees. However, traditional CP methods rely on the assumption of exchangeability among calibration samples, an assumption that is often violated in online environments, where data distributions are non-stationary (e.g., due to changing user behavior during a pandemic). To address the limitations posed by distributional shifts and exchangeability violations, numerous extensions to CP have been proposed [Tibshirani et al., 2019, Lei and Candès, 2021, Fannjiang et al., 2022, Barber et al., 2023, Plassier et al., 2024], incorporating techniques such as weighted quantiles, adaptive calibration, and distributional adjustment.

Among the earliest attempts to bring CP into an online setting, Gibbs and Candes [2021] proposed a method based on online convex optimization that adaptively adjusts prediction thresholds using per-instance coverage feedback. While this approach marked an important step toward online uncertainty quantification, it assumes access to true outcome labels at every time step to verify coverage, an assumption that is incompatible with privacy requirements in potentially sensitive user data such

---

[*]Equal contribution.
[†]Corresponding authors: Jinhan Xie<jinhanxie@ynu.edu.cn> and Ting Li<tingli@mail.shufe.edu.cn>.

39th Conference on Neural Information Processing Systems (NeurIPS 2025).

as patient records in healthcare or physiological signals from wearable devices. Moreover, the algorithm requires pre-specified learning rates, which may result in unstable prediction intervals that are either empty or unbounded in practice. Subsequent efforts have sought to mitigate these limitations by proposing more stable and adaptive procedures [Zaffran et al., 2022, Angelopoulos et al., 2023, Bhatnagar et al., 2023] . More recent work [Podkopaev et al., 2024, Ge et al., 2024, Angelopoulos et al., 2024, Gibbs and Candès, 2024] continues to advance online conformal prediction methodologies to better accommodate the challenges of non-stationary, real-time data streams.

Despite these important developments, existing approaches generally lack built-in privacy protection, leaving them vulnerable to information leakage in privacy-sensitive applications. As demonstrated by Angelopoulos et al. [2022], the conformal calibration step itself can leak private information unless explicitly privatized. This highlights the need for a principled framework that integrates uncertainty quantification with rigorous, end-to-end privacy guarantees. Differential Privacy (DP) [Dwork et al., 2006b,a], one of the most widely used frameworks for privacy-preserving data analysis, provides a rigorous mathematical definition that guarantees the output of a computation does not reveal sensitive information about any individual in the dataset. Due to its strong theoretical guarantees and practical utility, DP has seen widespread adoption across a variety of domains, including healthcare, wearable sensors, and public sector services [Dankar and El Emam, 2013, Li et al., 2022, Drechsler, 2023]. Recent research has focused extensively on the central differential privacy (CDP) model, which assumes the existence of a trusted server that can securely collect, store, and process raw user data. This framework has demonstrated effectiveness in a variety of applications, such as deep learning, federated learning, and synthetic data generation [Abadi et al., 2016, Adnan et al., 2022, Ponomareva et al., 2023, Olabim et al., 2024].

Unfortunately, the aforementioned methods based on the CDP model inherently depend on the assumption of trust in the central curator, making it vulnerable to technical breaches, insider misuse, and potential privacy violations if the central server is compromised. In contrast, the local differential privacy (LDP) model eliminates the need for a trusted aggregator by requiring that each user privatize their data at the source, before any transmission to a server or analyst [Kasiviswanathan et al., 2011, Duchi et al., 2018, He et al., 2023, Duchi and Ruan, 2024]. Owing to its strong privacy guarantees, LDP has been widely adopted in practice by major technology companies such as Google [Erlingsson et al., 2014, Song et al., 2021], Apple [Tang et al., 2017], and Meta [Yousefpour et al., 2021]. Despite its practical importance, the literature on CP under rigorous privacy guarantees remains relatively limited, even in offline settings. For example, Angelopoulos et al. [2022] proposed a method for constructing conformal prediction sets under CDP, but their approach is limited to static calibration datasets and centralized computation. Recent efforts have attempted to extend privacy guarantees to federated and decentralized frameworks. Humbert et al. [2023] applied the exponential mechanism within a federated learning paradigm to perform private quantile estimation across distributed clients. Similarly, Plassier et al. [2023] introduced noise into gradient updates to derive private prediction sets. Although these methods effectively enforce privacy during downstream processing, they fail to address a critical vulnerability in dynamic environments: the risk of exposure at the time of data acquisition. This highlights a critical gap in the existing literature, the lack of dynamic, model-free methods for constructing prediction sets under LDP that enable real-time, privacy-preserving decision-making.

The goal of this paper is to address the critical challenge outlined above. Specifically, we introduce an online CP framework under LDP constraints, a unified approach for distribution-free, privacy-preserving uncertainty quantification in streaming data environments. The overall flowchart of the proposed framework is illustrated in Figure 1. Our main contributions are summarized as follows:

- We propose a **trusted-curator-free** algorithm that operates fully online in a single-pass, under the **one-shot interaction model**, where each user is inquired exactly once. The algorithm leverages locally privatized binary feedback to ensure stringent privacy guarantees without direct access to raw user data. By design, it is both computationally and memory efficient, achieving **constant time and space complexity** ($\mathcal{O}(1)$ per instance), making it particularly well-suited for dynamic environments in both regression and classification tasks.

- Our framework is **model-agnostic**, supporting arbitrary black-box prediction models, including those trained under DP. In contrast to most existing methods, our approach provides the flexibility to construct fully end-to-end privacy-preserving pipelines when the underlying predictive model trained online is itself private.

- From a theoretical perspective, we provide a rigorous analysis of the algorithm's **LDP guarantees** and show that these extend to the entire prediction pipeline when the pre-trained model is also DP, thereby achieving full **end-to-end privacy protection**. In addition, we establish that the algorithm achieves **long-run coverage guarantees** at the nominal target level, regardless of the underlying predictive model.

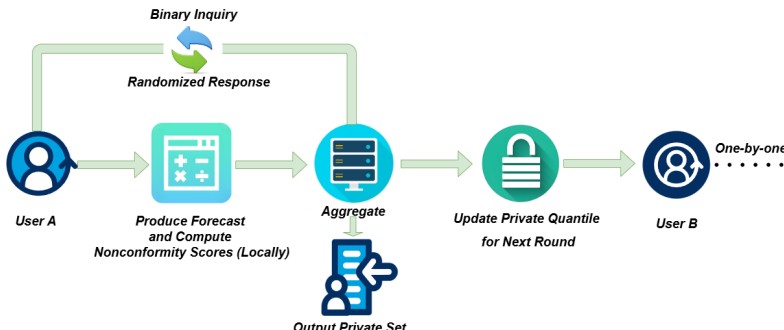

Figure 1: The server processes users sequentially in a one-by-one manner. Each user locally computes their own non-conformity score. Using the private quantile estimate from the previous round, the server constructs a prediction set for the current user and issues a binary inquiry. Upon receiving the user's randomized response, the server updates the private quantile estimate accordingly and proceeds to the next user in the sequence.

## 2 Preliminaries

### 2.1 Differential privacy

In this subsection, we review the basic concepts and some useful properties of CDP and LDP.

**Definition 2.1** (CDP, Dwork et al. [2006b]). *Let $\mathcal{X}$ be the sample space for an individual data, a randomized algorithm $M\colon \mathcal{X}^n \to \mathcal{R}$ is $(\epsilon, \delta)$-DP if and only if for every pair of adjacent datasets $X$, $X' \subset \mathcal{X}^n$ and for any measurable event $E \subseteq \mathcal{R}$, the inequality below holds:*

$$\Pr(M(X) \in E) \leq e^\epsilon \cdot \Pr(M(X') \in E) + \delta.$$

*When $\delta = 0$, then $M$ is called pure $\epsilon$-DP.*

This definition enforces privacy at the output level and assumes the existence of a trusted curator with access to the raw dataset. While such an assumption simplifies algorithm design, it does not eliminate the need for trust or mitigate the risks of internal data exposure. To address this limitation, LDP is given as follows, which shifts the privacy mechanism to the user side, requiring that data be randomized before collection, thereby removing the need for any trusted data aggregator. The formal definition of LDP is given below.

**Definition 2.2** (LDP, Xiong et al. [2020]). *A randomized algorithm $M : \mathcal{X} \to \mathcal{R}$ satisfies $(\epsilon, \delta)$-LDP if and only if, for any pair of input individual values $x, x' \in \mathcal{X}$, and for every measurable event $E \subseteq \mathcal{R}$, we have*

$$\Pr[M(x) \in E] \leq e^\epsilon \cdot \Pr[M(x') \in E] + \delta.$$

*When $\delta = 0$, this reduces to pure $\epsilon$-LDP.*

LDP ensures privacy at the individual level by requiring each user to randomize their data locally before sending it to the data collector. LDP supports different interaction models depending on how users communicate with the server. In this paper, we adopt the one-shot interactive LDP setting [Cheu et al., 2019, Liu et al., 2023], wherein each user applies a local randomizer exactly once and transmits privatized feedback to the server. This design substantially reduces communication overhead while providing rigorous privacy protection. Specifically, our algorithm assumes that each user is inquired only once, and the server updates its internal state solely based on the privatized responses it receives.

We now proceed to present several key properties of LDP that characterize how privacy guarantees are preserved under sequences of operations.

**Lemma 2.3.** *[Xiong et al., 2020]*

***Sequential Composition****: Suppose $n$ mechanisms $\{M_1, \ldots, M_n\}$ satisfy $\epsilon_i$-LDP, respectively, and are sequentially applied to the private data. Then, the combined mechanism formed by $(M_1, \ldots, M_n)$ in some order satisfies $(\sum_{i=1}^{n} \epsilon_i)$-LDP.*

***Parallel Composition****: Suppose $n$ mechanisms $\{M_1, \ldots, M_n\}$ satisfy $\epsilon_i$-LDP for $i = 1, 2, \ldots, n$, respectively, and are computed on a disjoint subset of the private data. Then, a mechanism formed by $(M_1(D_1), \ldots, M_n(D_n))$ satisfies $\max(\epsilon_i)$-LDP.*

***Postprocessing Property****: If mechanism $M_1$ satisfies $\epsilon$-LDP, then for any mechanism $M_2$, even if $M_2$ does not satisfy LDP, the composition $M_2(M_1(\cdot))$ also satisfies $\epsilon$-LDP.*

## 2.2 Problem formulation

In this paper, we consider an online learning framework in which data arrive sequentially as $\{(X_t, Y_t)\}_{t \geq 1}$, with $X_t \in \mathcal{X} \subset \mathbb{R}^d$ and $Y_t \in \mathcal{Y} \subset \mathbb{R}$. Classical online CP [Gibbs and Candes, 2021, Angelopoulos et al., 2023, Gibbs and Candès, 2024, Podkopaev et al., 2024] assumes the existence of a bounded non-conformity score function $S_t : \mathcal{X} \times \mathcal{Y} \to [0, D]$ at each time $t$, where the score $S_t = S_t(X_t, Y_t)$ quantifies the discrepancy between the model's prediction and the observed outcome. A common example in regression task is the absolute residual score, $S_t(X_t, Y_t) = |Y_t - \hat{f}_t(X_t)|$, where $\hat{f}_t : \mathcal{X} \to \mathbb{R}$ is the predictive model trained in an online manner. The primary goal is to construct a prediction set $\hat{C}_t$ at each time $t$ such that the long-run empirical coverage converges to a pre-specified nominal level $1 - \alpha$. Formally, we aim to satisfy:

$$\lim_{T \to \infty} \left| \frac{1}{T} \sum_{t=1}^{T} \mathbb{I}\left(Y_t \in \hat{C}_t(X_t)\right) - (1 - \alpha) \right| = 0. \tag{1}$$

where the prediction set is given by: $\hat{C}_t = \{y \in \mathcal{Y} : S_t(x, y) \leq q_t\}$, and the threshold $q_t$ is iteratively updated via $q_{t+1} = q_t + \eta_t(\mathbb{I}(Y_t \notin \hat{C}_t(X_t)) - \alpha)$, with $\eta_t$ denoting the step size. This update rule is interpreted as an online (sub)gradient descent algorithm on the quantile pinball loss:

$$q_{t+1} = q_t - \eta_t \partial \ell_{1-\alpha}(q_t, S_t) \tag{2}$$

with $\ell_{1-\alpha}(q, S_t) = (\mathbb{I}\{q \geq S_t\} - (1 - \alpha))(q - S_t)$.

While these methods achieve distribution-free coverage guarantees, they typically require access to accurate feedback indicating whether the true outcome $Y_t$ lies within the prediction set $\hat{C}_t$. In privacy-sensitive settings, this feedback can inadvertently disclose personal or confidential information, and users may be unwilling or unable to provide truthful responses. Additionally, the underlying predictive model $\hat{f}_t$ may itself be vulnerable to privacy attacks. This limitation highlights the need for a privacy-preserving online CP framework that mitigates information leakage while maintaining reliable online calibration.

# 3 Algorithm and main results

In this section, we systematically present our method for constructing online privacy-preserving CP sets using binary inquiries. As outlined in (2), each subgradient $\partial \ell_{1-\alpha}(q_t, S_t)$ is purely determined by the binary indicator variable representing whether the non-conformity score $S_t$ is less than or equal to the current quantile estimate $q_t$. This inherent binary structure enables us to implement a local randomization mechanism via randomized response, which directly supports LDP constraints. Motivated by this, we carefully design the binary inquiry presented to the user to adjust the stochastic gradient descent process via subgradient without violating privacy conditions. This leads to a modified update rule in which the deterministic subgradient is replaced with a privatized binary response, drawn from a locally randomized mechanism. Specifically, upon receiving the privatized binary feedback $L = \text{LRBR}(S_t, q_t, r)$, as defined in Algorithm 1, where $r \in (0, 1]$ is the probability of a truthful response, the subgradient $g_t$ is updated according to:

$$g_t = \begin{cases} 1 - (r(1-\alpha) + 0.5(1-r)), & \text{if } L = 1 \\ -(r(1-\alpha) + 0.5(1-r)), & \text{otherwise} \end{cases} \tag{3}$$

This formulation modulates the strength of privacy protection in the update step according to the response rate $r$: a smaller value of $r$ corresponds to stronger privacy protection, as it increases the amount of randomization in the binary feedback mechanism. Notably, in the special case where $r = 1$, the mechanism reduces to the non-DP setting, and the update $g_t$ degenerates to the standard subgradient $\partial \ell_{1-\alpha}(q_t, S_t)$, recovering the standard deterministic subgradient update without any privacy protection [Podkopaev et al., 2024]. The complete procedure is summarized in Algorithm 1.

---

**Algorithm 1:** Locally Randomized Binary Response (LRBR)

---

1 **Input:** local nonconformity score $S$, private quantile $q$, response rate $r$
2 $u \sim \text{Bernoulli}(r)$
3 $v \sim \text{Bernoulli}(0.5)$
4 **if** $u = 1$ **then**
5 $\quad$ **return** $1_{\{q > S\}}$
6 **else**
7 $\quad$ **return** $v$

---

In Algorithm 1, generating the random bit $v$ prior to the conditional branch may appear redundant; however, this design is intentional to prevent side-channel attacks, such as inferring the true value based on variations in response timing [Coppens et al., 2009, Lawson, 2009].

To enable private, adaptive quantile estimation in an online setting, we adopt a coin betting-based online convex optimization framework [Orabona and Pal, 2016, Cutkosky and Orabona, 2018, Podkopaev et al., 2024]. This parameter-free and robust method reformulates the learning process as a repeated betting game framework, providing regret guarantees even in adversarial environments. Within this framework, a gambler sequentially bets a fraction $\lambda_t \in [-1, 1]$ of their current wealth $W_{t-1}$ on an outcome $c_t \in [-1, 1]$, which may be adversarially chosen as in [Orabona and Pal, 2016]. Starting from an initial endowment $W_0 > 0$, the cumulative wealth evolves according to: $W_t = W_0 + \sum_{i=1}^{t} \lambda_i W_{i-1} c_i$. To adaptively adjust $\lambda_t$ based on sequential feedback, we use the Krichevsky-Trofimov (KT) estimator [Krichevsky and Trofimov, 1981], which sets: $\lambda_t = \sum_{i=1}^{t-1} c_i / t$. To integrate this with our privatized quantile estimation, we define the feedback as $c_t := -g_t$, where $g_t$ is the privatized subgradient in equation (3). The signed bet $w_t := \lambda_t W_{t-1}$ is then interpreted as the current privatized estimate of the $(1 - \alpha)$-quantile, denoted as $q_t$. This feedback is used to update the coin betting procedure at each step. The resulting design enables the algorithm to adaptively refine its quantile estimate based on privatized, binary feedback while maintaining rigorous LDP guarantees. We summarize the proposed procedure for the regression task in Algorithm 2. The corresponding extension to classification tasks is presented in Section 4.

In our framework, the predicted value $\hat{Y}_t$ will be transmitted to the curator in order to construct the prediction interval $\hat{C}_t = [\hat{Y}_t - q_t, \hat{Y}_t + q_t]$. As such intervals inherently disclose the quantile threshold $q_t$, inquiring the user based on this value does not introduce additional privacy risk. Importantly, $q_t$ is updated via the above proposed mechanism that satisfies LDP, thereby ensuring rigorous individual-level privacy protection, even when $q_t$ is externally observable. Specifically, $q_t$ is estimated via binary inquiries, which provide several practical advantages. Each inquiry conveys only a single bit of information, substantially reducing communication overhead and improving transmission efficiency. In contrast to open-ended inquiries, binary inquiries are more intuitive and cognitively accessible, thereby enhancing user comprehension and response fidelity [Brown et al., 1996]. Beyond communication efficiency, our framework facilitates fully online computation, eliminating the need to store or re-access historical data. As a result, it significantly reduces memory and computational demands, making our method particularly well-suited for streaming applications [Lee et al., 2022].

In online learning, regret quantifies the cumulative difference between the learner's performance and that of a fixed comparator. Formally, let $\{q_t\}_{t=1}^{T}$ denote the sequence of decisions made by the algorithm. The regret with respect to a fixed comparator $q^{\star} \in \mathbb{R}$ is given by:

$$R_T(q^{\star}) := \sum_{t=1}^{T} \ell_t(q_t) - \ell_t(q^{\star}).$$

We provide the regret bound as follows in Corollary 3.1:

**Algorithm 2:** Binary Private Online Conformal Prediction

---

**Input:** Data stream $\{(X_t, Y_t)\}_{t\geq 1}$; Response rate $r_t \in (0,1)$; Miscoverage level $\alpha \in (0,1)$;
Initialize $W_0 = 1$, $\lambda_1 = 0$, $q_1 = 0$

**1  for** $t = 1, 2, \ldots$ **do**
      `// Train a predictive model` $\hat{f}_t(\cdot)$
**2**     Predict $\hat{Y}_t \leftarrow \hat{f}_t(X_t)$;
      `// Generate conformal prediction interval`
      **Output:** the private prediction interval: $\hat{C}_t := [\hat{Y}_t - q_t, \hat{Y}_t + q_t]$;
      `// Compute nonconformity score (kept locally)`
**3**     Nonconformity score: $S_t = |Y_t - \hat{Y}_t|$;
      `// Preparing updated rule for next round`
**4**     Inquiry LRBR response: $L \leftarrow \text{LRBR}(S_t, q_t, r_t)$;
**5**     **if** $L = 1$ **then**
**6**        $\big\lfloor$ $g_t \leftarrow 1 - (r_t(1-\alpha) + 0.5(1-r_t))$;
**7**     **else**
**8**        $\big\lfloor$ $g_t \leftarrow -(r_t(1-\alpha) + 0.5(1-r_t))$;
**9**     Update: $W_t \leftarrow W_{t-1} - g_t q_t$;
**10**    Update: $\lambda_{t+1} \leftarrow \frac{t}{t+1}\lambda_t - \frac{1}{t+1}g_t$;
**11**    Update: $q_{t+1} \leftarrow \lambda_{t+1}W_t$;

---

**Corollary 3.1.** *Let $\{q_t\}_{t=1}^{T} \subseteq \mathbb{R}$ denote the sequence of decisions produced by the proposed algorithm. Under the coin betting framework with the KT potential, the cumulative regret with respect to any fixed comparator $q^\star \in \mathbb{R}$ satisfies the following bound:*

$$R_T(q^\star) := \sum_{i=1}^{T} \ell_t(q_t) - \ell_t(q^\star) \leq 1 + |q^\star| \cdot \sqrt{2T \ln(1 + CT|q^\star|)} - \frac{1}{e\sqrt{\pi T}},$$

*where $C > 0$ is a universal constant, and $e$ denotes the base of the natural logarithm.*

Corollary 3.1 establishes a sublinear cumulative regret bound within the coin betting framework. Specifically, the regret bound of order $O(\sqrt{T \ln T})$ ensures that the cumulative loss incurred by the algorithm relative to any fixed comparator $q^\star$ grows sublinearly over time. Consequently, the average per-round regret asymptotically converges to zero as $T \to \infty$.

**Theorem 3.2.** *Algorithm 1 is an $(\epsilon, 0)$-randomizer with $\epsilon = \log\left((1+r)/(1-r)\right)$.*

*Proof.* With probability $r$, the user returns a truthful binary response indicating whether $S_t < q_t$; otherwise, a fair coin is flipped. This randomized mechanism masks the true response, and the privacy level $\epsilon$ is determined by the worst-case distinguishability between outputs for different true answers. A full derivation is deferred to Appendix C.2. $\qquad\square$

**Theorem 3.3.** *Algorithm 2 satisfies $(\max_{1\leq j\leq t} \epsilon_j, 0)$-LDP, where $t \geq 1$.*

*Proof.* By parallel composition and the post-processing property, the guarantee holds. A detailed argument is deferred to Appendix C.3. $\qquad\square$

Theorems 3.2 and 3.3 establish that the proposed algorithms satisfy rigorous LDP guarantees. The proposed algorithms ensure that individual-level privacy is preserved at every iteration, without the need to store or re-access raw data, thereby enabling both privacy protection and memory efficiency in streaming environments.

Although our algorithm satisfies LDP at the user level, an adversary could still exploit vulnerabilities in the underlying predictive model to infer sensitive information. To address this concern, we establish a rigorous form of privacy protection: when the predictive model is dynamically trained under LDP constraints, our framework also ensures end-to-end privacy guarantees across the entire prediction pipeline.

**Theorem 3.4.** *Suppose the predictive model satisfies* $(\max_{1 \leq j \leq t} \varepsilon_j, \max_{1 \leq j \leq t} \delta_j)$*-LDP. The entire pipeline, mapping private individual data to the final prediction interval $\hat{C}_t$ inherits LDP guarantees. Then, Algorithm 2 ensures* $(\max_{1 \leq j \leq t}(\epsilon_j + \varepsilon_j), \max_{1 \leq j \leq t} \delta_j)$*-LDP, where $t \geq 1$.*

*Proof.* By additionally applying the sequential composition property, the theorem follows. The complete proof is deferred to Appendix C.4. □

We illustrate this process in Figure 13, which empirically validates the consistency between our simulation results and the theoretical guarantee established in Theorem 3.4.

**Theorem 3.5.** *Let the target miscoverage level $\alpha \in (0, 1/2)$ be fixed, and let $r \in (0, 1]$ denote the probability of a truthful response. Suppose that the non-conformity scores are bounded, such that $S_t \in [0, D]$ for all $t = 1, 2, \ldots$, where $D > 0$ is a finite constant. Then, the proposed procedure described in Algorithm 2 satisfies the following long-run coverage guarantee:*

$$\lim_{T \to \infty} \left| \frac{1}{T} \sum_{t=1}^{T} \mathbb{I}\left(Y_t \in \hat{C}_t\right) - (1 - \alpha) \right| = 0. \tag{4}$$

*Proof.* Although each feedback bit is randomized, its expectation coincides with the true subgradient:

$$\mathbb{E}[g_t] = r \cdot \left(\mathbb{I}\{S_t < q_t\} - (1 - \alpha)\right).$$

Hence, the update rule drives $q_t$ toward the $(1 - \alpha)$ quantile in expectation. By boundedness of the scores and stability of the coin-betting update, the prediction sets converge to the desired long-run coverage. A complete derivation is provided in Appendix C.5. □

Theorem 3.5 establishes that the proposed method achieves valid long-run coverage guarantees that are independent of the underlying predictive model, regardless of whether the model is dynamically trained with or without privacy constraints. However, because $q_t$ is defined as the $(1 - \alpha)$-quantile of the non-conformity scores, models trained under LDP often exhibit decreased predictive accuracy due to the injected noise. This typically results in larger non-conformity scores and consequently wider prediction intervals. To illustrate this effect, we include a representative experiment using the privately trained predictive model(PTM) from Abadi et al. [2016], dynamically trained under LDP constraints; see Figure 12.

## 4 Extension to classification tasks

In this section, we extend our framework from regression to multi-class classification problems, where the response variable takes values in a finite label set $\mathcal{Y}$. Unlike regression, where prediction intervals are centered around scalar estimates, classification requires constructing prediction sets containing likely class labels. This shift necessitates adapting both the definition of non-conformity scores and the structure of the prediction region. Given a probabilistic classifier $\hat{f} : \mathbb{R}^d \to [0, 1]^{|\mathcal{Y}|}$, we define the non-conformity score for a candidate label $y \in \mathcal{Y}$ as

$$S_{\hat{f}}(X, y) = 1 - \hat{p}_y(X),$$

where $\hat{p}_y(X)$ denotes the predicted probability assigned to class $y$, typically computed via softmax over the model's logits. Intuitively, this score captures the model's uncertainty about the correctness of label $y$: smaller scores correspond to more confident predictions. At each time step $t$, the prediction set is constructed by including all labels whose non-conformity scores fall below a dynamic quantile threshold:

$$\hat{\Gamma}_t(X_t) = \{y \in \mathcal{Y} \mid S_{\hat{f}}(X_t, y) \leq q_t\}.$$

The threshold $q_t$, analogous to the conformal quantile used in the regression setting, is updated sequentially based on privatized binary feedback. To ensure privacy, we adopt the same binary inquiry strategy as in Section 3. Specifically, the non-conformity score for the true label, $S_t = S_{\hat{f}}(X_t, Y_t)$, is compared against the current threshold $q_t$ and the result of this comparison is encoded as a binary signal, which is privatized using a local binary randomized mechanism. The privatized feedback is then used to compute a subgradient $g_t$ (as defined in Equation (3)), and the sign-reversed quantity $c_t := -g_t$ is passed to the coin-betting algorithm to update $q_t$ in an efficient and privacy-preserving manner. The complete procedure for the classification task is summarized in Algorithm 3.

Table 1: Coverages and widths for the proposed method, DPCP and DtACI in Example 5.1. Numbers in parentheses denote standard deviations for coverage ($\times 10^{-2}$) and for width ($\times 10^{-1}$).

| Method | no-privacy | | $\epsilon = 3$ | | $\epsilon = 1$ | | $\epsilon = 0.5$ | |
|---|---|---|---|---|---|---|---|---|
| | Coverage | Width | Coverage | Width | Coverage | Width | Coverage | Width |
| **Case A** | | | | | | | | |
| Proposed | 0.890 (0.03) | 3.43 (0.30) | 0.889 (0.30) | 3.42 (0.40) | 0.875 (1.00) | 3.36 (1.20) | 0.853 (2.10) | 3.28 (2.70) |
| DPCP | 0.900 (0.51) | 8.29 (1.02) | 0.904 (0.52) | 8.40 (1.07) | 0.911 (0.61) | 8.60 (1.48) | 0.922 (0.90) | 8.95 (2.96) |
| DtACI | 0.897 (0.09) | 3.47 (0.30) | * | * | * | * | * | * |
| **Case B** | | | | | | | | |
| Proposed | 0.890 (0.05) | 4.46 (0.96) | 0.889 (0.26) | 4.44 (1.30) | 0.874 (0.90) | 4.22 (2.91) | 0.850 (1.82) | 3.83 (5.06) |
| DPCP | 0.900 (0.54) | 9.00 (1.45) | 0.904 (0.53) | 9.14 (1.56) | 0.911 (0.60) | 9.40 (2.19) | 0.923 (0.93) | 9.89 (4.55) |
| DtACI | 0.899 (0.07) | 4.76 (1.02) | * | * | * | * | * | * |
| **Case C** | | | | | | | | |
| Proposed | 0.890 (0.03) | 3.26 (0.30) | 0.889 (0.27) | 3.26 (0.37) | 0.875 (1.06) | 3.21 (1.03) | 0.852 (1.92) | 3.11 (1.89) |
| DPCP | 0.900 (0.50) | 4.43 (0.63) | 0.904 (0.51) | 4.49 (0.60) | 0.911 (0.61) | 4.60 (0.88) | 0.920 (1.01) | 4.80 (1.99) |
| DtACI | 0.899 (0.07) | 3.34 (0.27) | * | * | * | * | * | * |
| **Case D** | | | | | | | | |
| Proposed | 0.890 (0.03) | 3.26 (0.31) | 0.889 (0.22) | 3.26 (0.36) | 0.875 (1.02) | 3.21 (1.02) | 0.853 (1.90) | 3.11 (2.11) |
| DPCP | 0.900 (0.53) | 3.29 (0.41) | 0.904 (0.53) | 3.33 (0.42) | 0.911 (0.62) | 3.40 (0.58) | 0.922 (0.97) | 3.53 (1.19) |
| DtACI | 0.899 (0.08) | 3.34 (0.29) | * | * | * | * | * | * |

## 5 Experiments

We evaluate the finite-sample performance of the proposed estimator by comparing it with two baselines: DPCP [Angelopoulos et al., 2022], an offline private conformal method, and DtACI [Gibbs and Candès, 2024], an online adaptive method using expert aggregation. For ease of interpretation, the privacy parameter is set to $\epsilon = 0.5, 1, 2, 3$, with the corresponding values of $r$ provided in Table 4. We define the miscoverage level as $\alpha = 0.1$.

**Example 5.1** (**Regression on Synthetic Data**). In this example, we evaluate the proposed method on the regression task using synthetic data. We generate data similar to Barber et al. [2023] via $x_t \sim \mathcal{N}(0, I_5)$ and $y_t = x_t^\top \beta_t + \varepsilon_t$ for $t = 1, \ldots, 10,000$, where $\beta_t \in \mathbb{R}^5$ and $\varepsilon_t$ is from a normal distribution and independent of $x_t$. We consider four cases. (A) **Abrupt shifts with homoskedastic errors**; (B) **Abrupt shifts with heteroskedastic errors**; (C) **Smooth shifts with homoskedastic errors**; (D) **No shifts with homoskedastic errors** The sample size is set to 10,000. Long-run coverage and interval width are evaluated over these data points and averaged across 200 simulation runs. See Appendix B.1 for data generation.

Table 1 reports coverage and interval width for both methods across four cases at the 90% confidence level. In drifting environments that are closer to real-world data streams, the proposed method achieves slightly lower but comparable coverage to DPCP, while producing substantially narrower intervals. Specifically, DPCP's widths more than double in Cases A and B and increase by over 40% in Case C. This is expected, as DPCP operates in a fixed-sample offline setting and constructs conservative intervals without adapting to distributional shifts. In contrast, our online method updates intervals in real time, allowing for more efficient and adaptive inference. In the static setting without distributional shifts, as in Case D, DPCP's higher coverage becomes more favorable. For DtACI, which is designed for online settings without privacy constraints, we compare it under the no-privacy scenario. The proposed method again yields shorter intervals while maintaining comparable performance, with slightly lower coverage.

We compare the computational efficiency of the three methods. Table 2 shows that DtACI, with six experts, requires approximately 100 times more computation time due to its complex update rules, and DPCP requires approximately 10 times.

The left panel of Figure 2 shows instant coverage, instant width, long-run coverage, and long-run width for DtACI with initialization values $\alpha_1 \in \{0.1, 0.9\}$, and for the proposed method under varying privacy levels. DPCP is omitted as it is an offline method. All methods exhibit coverage oscillations around the target level of $1 - \alpha$, with noticeable drops at change points but rapid recovery

thereafter. Lower $\epsilon$ values result in slightly larger fluctuations. Initially, smaller $\epsilon$ values lead to lower long-run coverage, but all methods eventually converge to the target level. A similar pattern is observed for instant width and long-run width. The results also show that DtACI is sensitive to initialization and requires each expert to adjust predictions based on interval coverage, leading to frequent feedback requests. As the number of experts increases, this can cause user fatigue and raise privacy concerns, limiting its use in privacy-sensitive settings. In contrast, the right panel of Figure 2 shows that the proposed method is robust to initialization, demonstrating consistent performance across different settings. Results for Case B in Figure 4, Case C in Figure 5, and Case D in Figure 6 of the Appendix exhibit similar patterns.

Table 2: Computational time (in seconds) of the proposed method, DPCP and DtACI.

| Method | T=5000 | T=10000 | T=15000 | T=20000 | T=25000 |
|---|---|---|---|---|---|
| DtACI | 3.46 | 7.03 | 11.17 | 14.84 | 19.34 |
| DPCP | 0.931 | 1.003 | 1.077 | 1.234 | 1.378 |
| Proposed | 0.037 | 0.069 | 0.100 | 0.133 | 0.167 |

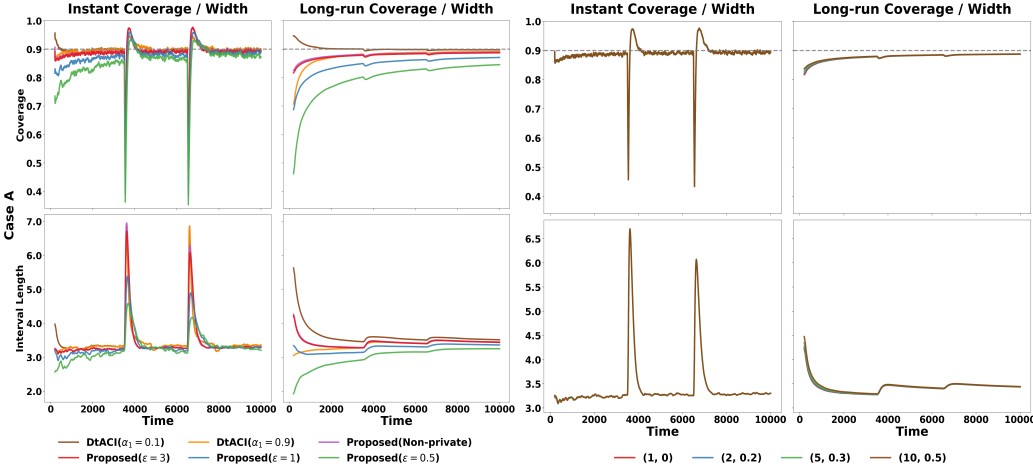

Figure 2: Results for Case A in Example 5.1. Left panels: comparison of methods by instant coverage, instant width, long-run coverage, and long-run width. Right panels: impact of initialization settings $(W_0, \lambda_1)$ for the proposed method with $\epsilon = 3$.

.

**Example 5.2** (**Classification on synthetic data**). We also conduct a classification study using synthetic data, encompassing three distribution shift scenarios plus one no shift environment. The findings are generally similar to those in the regression setting. To save space, we relegate the detailed results to Appendix B.2.

**Example 5.3** (**Regression on Real Data**). We use the ELEC2 dataset [Harries et al., 1999]. As shown in Figure 11 of the Appendix, lower $\epsilon$ values lead to slightly larger fluctuations in coverage. Nevertheless, the long-run coverage stabilizes around $1 - \alpha$ within the first quarter of the data stream for all methods. Similar patterns are observed in the width and long-run width. Further details are provided in Appendix B.3.

**Example 5.4** (**Classification on Real Data**). We assess the classification performance of our method using the real WISDM dataset [Kwapisz et al., 2011], a benchmark for smartphone-based activity recognition. We focus on two representative users — User 10 and User 14 — who completed 5 and 4 activity types, respectively, forming two independent tasks. For each task, we construct a time-indexed data stream and implement the model $\hat{f}_t$ using XGBoost. Rolling coverage is calculated using a sliding window of 200 points to capture short-term fluctuations, while long-run coverage reflects the cumulative average over all time steps. Set size [Angelopoulos et al., 2022] indicates the number of classes included in the prediction set, representing model uncertainty. Lower values at a given coverage level indicate better performance.

Figure 3 shows the results for both users under varying privacy settings of DtACI and the proposed method. Patterns are generally consistent with the simulation studies. With stronger privacy ($\epsilon \to 1$), rolling coverage shows increased variability due to amplified noise. Long-run coverage initially decreases but gradually converges to the target level. For set size, lower $\epsilon$ values slightly increase the occurrence of larger sets, indicating increased uncertainty. However, small and medium-sized sets remain dominant overall. DtACI is also sensitive to the initial setting of $\alpha_1$, initially deviating more but ultimately aligning with the non-private and low-noise settings.

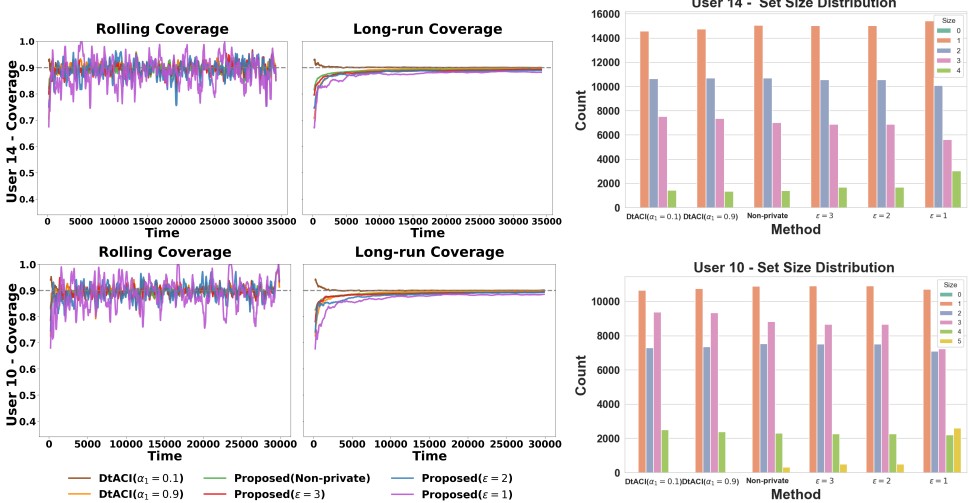

Figure 3: Evaluation of method performance in terms of coverage and set size for User 14 (top), and User 10 (bottom).

# 6    Conclusion and Limitations

This paper proposes a novel and practical framework for online conformal prediction under LDP. By combining randomized binary feedback with a coin-betting update scheme, the method enables one-pass, model-free prediction set construction with formal privacy guarantees. It supports both regression and classification, and operates efficiently in streaming settings with constant memory. Despite its strengths, several limitations remain. First, Theorem 3.5 provides only asymptotic coverage guarantees; deriving non-asymptotic bounds under LDP is an important direction for future work. Second, our approach employs standard nonconformity scores, which are broadly applicable but may yield conservative prediction sets. Incorporating ideas from recent methods such as CP-Gen [Bai et al., 2022], which learn data-adaptive scores via constrained ERM, could lead to more efficient prediction sets while preserving valid coverage. Another promising direction is to extend our framework to multivariate private online conformal prediction, building on recent advances such as [Xu et al., 2024] and [Braun et al., 2025]. Such extensions would broaden the applicability of private online conformal prediction to high-dimensional and structured data streams.

## Acknowledgements

We sincerely thank the reviewers, ACs, SACs, and PCs for their time, constructive feedback, and thoughtful discussions. Qiangqiang Zhang and Xinwei Feng were supported by the National Key R&D Program of China (No. 2023YFA1008701) and the National Natural Science Foundation of China (Grant Nos. 12371148, 12326603, and 12431017). Chenfei Gu's research was supported by the Fundamental Research Funds for the Central Universities (No. CXJJ-2024-448). Jinhan Xie's research was partially supported by the National Key R&D Program of China (102022YFA1003701) and the National Natural Science Foundation of China (No.12501388). Ting Li's research was partially supported by the National Natural Science Foundation of China (No.12571304), the Shanghai Pujiang Programme (No. 24PJC030), CCF- DiDi GAIA Collaborative Research Funds and the Program for Innovative Research Team of Shanghai University of Finance and Economics.

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

# Appendix

## A  Algorithm for classification under binary private online conformal prediction

---

**Algorithm 3:** Binary Private Online Conformal Prediction (Classification)

---

**Input:** Data stream $\{(X_t, Y_t)\}_{t \geq 1}$; Response rate $r_t \in (0, 1)$; Miscoverage level $\alpha \in (0, 1)$;
Initialize $W_0 = 1$, $\lambda_1 = 0$, $q_1 = 0$

1 **for** $t = 1, 2, \ldots$ **do**

    // Train a predictive model $\hat{f}_t(\cdot)$

2    Compute probability vector: $\hat{p}_t(X_t) \leftarrow \hat{f}_t(X_t)$, where $\hat{p}_t(X_t) = [\hat{p}_{1,t}(X_t), \ldots, \hat{p}_{|\mathcal{Y}|,t}(X_t)]$;

    // Generate conformal prediction set

    **Output:** the private prediction set: $\hat{\Gamma}_t(X_t) = \{y \in \mathcal{Y} \mid 1 - \hat{p}_{y,t}(X_t) \leq q_t\}$;

    // Calculate nonconformity score (kept locally)

3    Nonconformity score: $S_t = 1 - \hat{p}_{Y_t,t}(X_t)$;

    // Preparing updated rule for next round

4    Inquiry LRBR response: $L \leftarrow \text{LRBR}(S_t, q_t, r_t)$;

5    **if** $L = 1$ **then**

6        $g_t \leftarrow 1 - (r_t(1 - \alpha) + 0.5(1 - r_t))$;

7    **else**

8        $g_t \leftarrow -(r_t(1 - \alpha) + 0.5(1 - r_t))$;

    // Update wealth and quantile estimate

9    Update: $W_t \leftarrow W_{t-1} - g_t q_t$;

10    Update: $\lambda_{t+1} \leftarrow \frac{t}{t+1} \lambda_t - \frac{1}{t+1} g_t$;

11    Update: $q_{t+1} \leftarrow \lambda_{t+1} W_t$;

---

## B  Additional Results

### B.1  Regression Setting

**Data generation for regression.**  Similar to Barber et al. [2023], we generate data via $x_t \sim \mathcal{N}(0, I_5)$ and $y_t = x_t^\top \beta_t + \varepsilon_t$ for $t = 1, \ldots, 10{,}000$, where $\beta_t \in \mathbb{R}^5$ and $\varepsilon_t$ is a Gaussian noise. We consider the following four scenarios:

**(1) Abrupt regime shifts.** We define three fixed coefficient vectors:

$$\beta^{(1)} = (1, 2, 1, 0, 0), \quad \beta^{(2)} = (0, -1, -2, -1, 0), \quad \beta^{(3)} = (0, 0, 1, 2, 1),$$

and assign $\beta_t = \beta^{(j)}$ for three consecutive segments of $t$. Under this setting, we consider two noise variants: **Case (A)** uses homoskedastic noise $\varepsilon_t \sim \mathcal{N}(0, 1)$, and **Case (B)** uses heteroskedastic noise $\varepsilon_t = x_{t,1}^2 \cdot \eta_t$ with $\eta_t \sim \mathcal{N}(0, 1)$.

**(2) Smooth concept drift.** We let $\beta_t$ evolve linearly over time:

$$\beta_t = (1 - \alpha_t)\beta_{\text{start}} + \alpha_t \beta_{\text{end}}, \quad \alpha_t = \frac{t - 1}{n - 1},$$

where $\beta_{\text{start}} = (1, 2, 1, 0, 0)$ and $\beta_{\text{end}} = (0, 0, 1, 2, 1)$. The noise is homoskedastic with $\varepsilon_t \sim \mathcal{N}(0, 1)$, denoted as **Case (C)**.

**(3) Fixed environment.** We keep the coefficient vector fixed throughout the whole stream:

$$\beta_t = (1, 2, 1, 0, 0), \quad \forall t = 1, \ldots, 10{,}000,$$

with homoskedastic noise $\varepsilon_t \sim \mathcal{N}(0,1)$. This is denoted as **Case (D)**.

Figures 4-5 present the simulation results for Case B and Case C, respectively. The general characteristics of the proposed method are consistent with those observed in Case A, though some differences arise due to the distinct data generation settings. In Case B, the heteroskedastic noise structure leads to more pronounced fluctuations and wider intervals compared to Case A. Conversely, in Case C and Case D, the absence of abrupt regime shifts results in relatively stable coverage and interval width throughout the data stream.

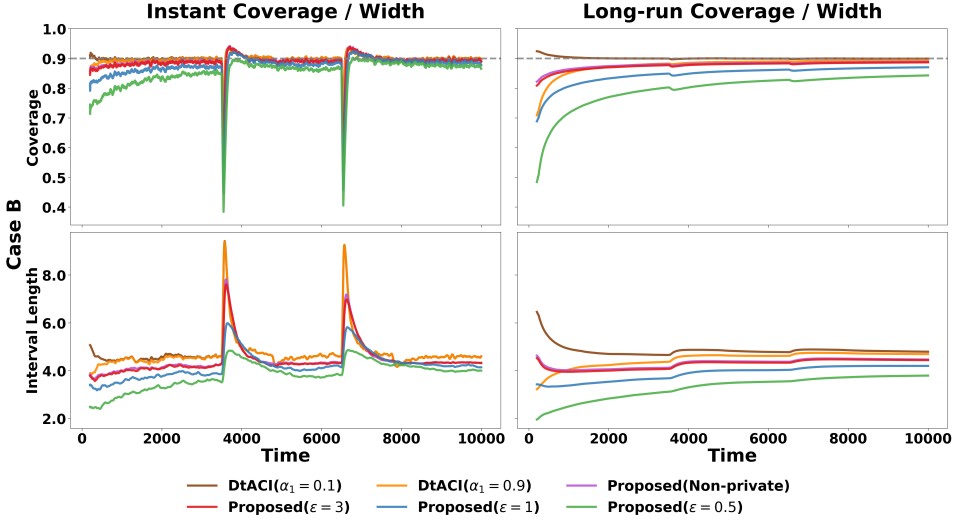

Figure 4: Simulation results for regression in Case B, comparing different methods in terms of instant coverage, instant width, long-run coverage, and long-run width. The results are averaged over 200 runs, excluding the first 200 data points.

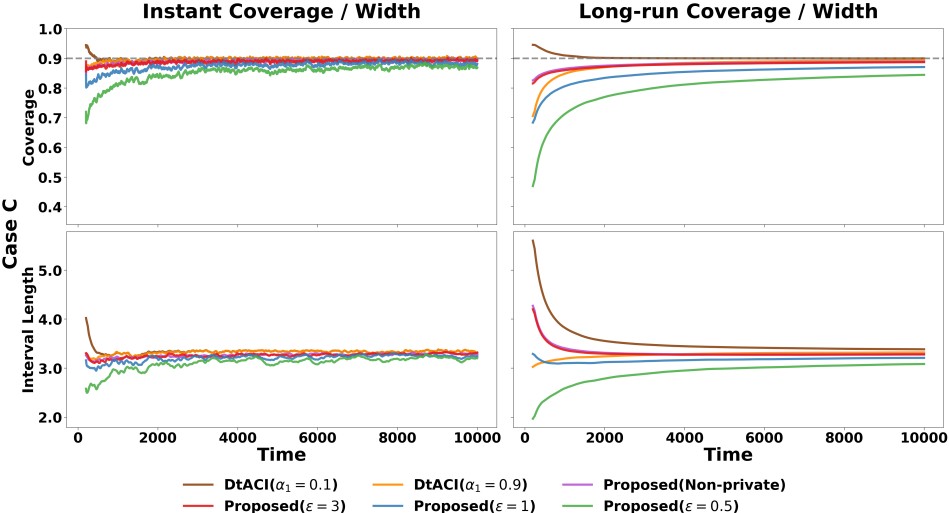

Figure 5: Simulation results for regression in Case C, comparing different methods in terms of instant coverage, instant width, long-run coverage, and long-run width. The results are averaged over 200 runs, excluding the first 200 data points.

## B.2   Classification Setting

**Data generation for classification.**   We evaluate our method under four scenarios, each generating a data stream $\{(x_t, y_t)\}_{t=1}^T$, where $x_t \sim \mathcal{N}(0, I_p)$ and $y_t \in \{0, \dots, K-1\}$ is drawn from the softmax

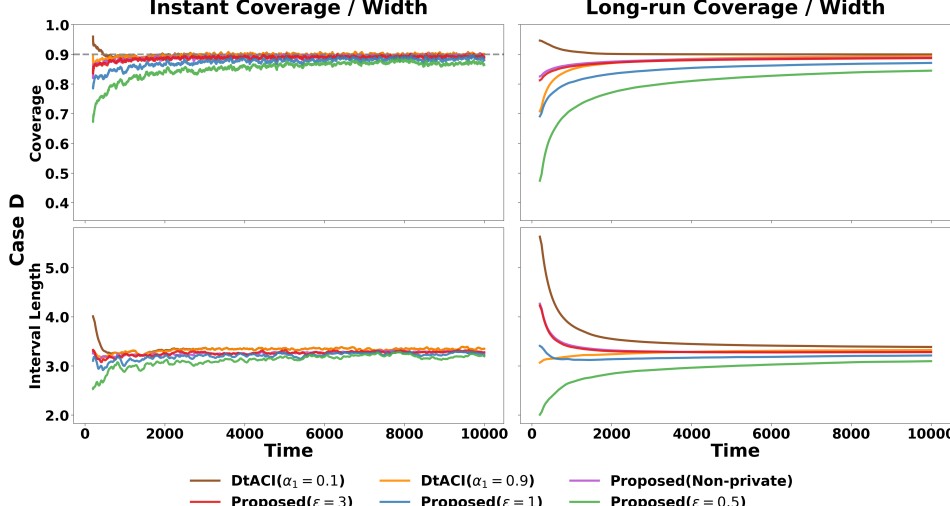

Figure 6: Simulation results for regression in Case D, comparing different methods in terms of instant coverage, instant width, long-run coverage, and long-run width. The results are averaged over 200 runs, excluding the first 200 data points.

model:

$$\mathbb{P}(y_t = k \mid x_t) = \frac{\exp(\langle \beta_t^{(k)}, x_t \rangle)}{\sum_{j=0}^{K-1} \exp(\langle \beta_t^{(j)}, x_t \rangle)}.$$

For each scenario, the coefficient vectors $\beta_t^{(k)}$ evolve linearly over time as:

$$\beta_t^{(k)} = (1 - \alpha_t)\beta_{\text{start}}^{(k)} + \alpha_t \beta_{\text{end}}^{(k)}, \quad \alpha_t = \frac{t-1}{T-1}.$$

**Smooth Drift (Case 1)**   In this setting, we consider $K = 3$ classes and $p = 3$ features. Classes 0 and 1 drift symmetrically along the first feature, while class 2 remains fixed. The coefficient vectors are defined as follows:

$$\beta_{\text{start}}^{(0)} = [-1, 0, 0], \quad \beta_{\text{end}}^{(0)} = [1, 0, 0], \quad \beta_{\text{start}}^{(1)} = [1, 0, 0], \quad \beta_{\text{end}}^{(1)} = [-1, 0, 0], \quad \beta_t^{(2)} \equiv [0, 0, 1].$$

**Amplified Drift (Case 2)**   This setting is a variant of the Smooth Drift scenario but with doubled drift magnitude. The number of classes and features remains the same ($K = 3, p = 3$). The coefficient vectors are defined as:

$$\beta_{\text{start}}^{(0)} = [-2, 0, 0], \quad \beta_{\text{end}}^{(0)} = [2, 0, 0], \quad \beta_{\text{start}}^{(1)} = [2, 0, 0], \quad \beta_{\text{end}}^{(1)} = [-2, 0, 0], \quad \beta_t^{(2)} \equiv [0, 0, 2].$$

**Class Emergence (Case 3)**   This setting introduces a new class that gradually emerges by activating a specific feature, while the other classes remain fixed. Here, we consider $K = 4$ classes and $p = 5$ features. The coefficient vectors are defined as:

$$\beta_t^{(0)} \equiv [2, 0, 0, 0, 0], \quad \beta_t^{(1)} \equiv [-2, 0, 0, 0, 0], \quad \beta_t^{(2)} \equiv [0, 0, 2, 0, 0],$$
$$\beta_{\text{start}}^{(3)} = [0, 0, 0, 0, 0], \quad \beta_{\text{end}}^{(3)} = [0, 0, 0, 0, 4].$$

**No Drift (Case 4)**   As a control setting, we consider no drift: all classes remain fixed ($K = 3, p = 3$):

$$\beta_t^{(0)} \equiv [-1, 0, 0], \quad \beta_t^{(1)} \equiv [1, 0, 0], \quad \beta_t^{(2)} \equiv [0, 0, 1].$$

Figure 7 shows the coverage and set size of the prediction sets for the outcome labels. For a fixed coverage level, a smaller set size indicates higher efficiency. The observed patterns align closely with those in the regression setting. For DtACI, initialization is particularly sensitive. When the initial $\alpha_1$ is set equal to the target miscoverage level $\alpha = 0.1$, the initial prediction sets are relatively large and

Table 3: Coverages and set sizes for the proposed method , DPCP and DtACI in Example 5.2. Numbers in parentheses denote standard deviations for coverage ($\times 10^{-2}$) and for width ($\times 10^{-1}$).

| Method | no-privacy | | $\epsilon = 3$ | | $\epsilon = 1$ | | $\epsilon = 0.5$ | |
|---|---|---|---|---|---|---|---|---|
| | Coverage | size | Coverage | size | Coverage | size | Coverage | size |
| **Case 1** | | | | | | | | |
| Proposed | 0.890 (0.02) | 2.17 (0.16) | 0.889 (0.24) | 2.17 (0.20) | 0.875 (1.05) | 2.16 (0.55) | 0.854 (2.09) | 2.11 (1.02) |
| DPCP | 0.900 (0.42) | 2.19 (0.14) | 0.902 (0.41) | 2.20 (0.15) | 0.905 (0.44) | 2.22 (0.17) | 0.911 (0.53) | 2.25 (0.24) |
| DtACI | 0.901 (0.08) | 2.20 (0.15) | * | * | * | * | * | * |
| **Case 2** | | | | | | | | |
| Proposed | 0.890 (0.03) | 1.69 (0.16) | 0.889 (0.25) | 1.69 (0.22) | 0.875 (1.04) | 1.70 (0.62) | 0.853 (2.03) | 1.67 (1.16) |
| DPCP | 0.901 (0.42) | 1.72 (0.16) | 0.902 (0.42) | 1.73 (0.16) | 0.906 (0.44) | 1.74 (0.18) | 0.911 (0.52) | 1.77 (0.23) |
| DtACI | 0.901 (0.08) | 1.73 (0.16) | * | * | * | * | * | * |
| **Case 3** | | | | | | | | |
| Proposed | 0.890 (0.02) | 1.71 (0.19) | 0.889 (0.23) | 1.72 (0.25) | 0.875 (1.04) | 1.77 (0.94) | 0.852 (1.86) | 1.75 (1.62) |
| DPCP | 0.900 (0.42) | 1.74 (0.19) | 0.901 (0.42) | 1.75 (0.19) | 0.905 (0.45) | 1.78 (0.22) | 0.910 (0.54) | 1.81 (0.32) |
| DtACI | 0.900 (0.08) | 1.75 (0.19) | * | * | * | * | * | * |
| **Case 4** | | | | | | | | |
| Proposed | 0.890 (0.02) | 1.92 (0.17) | 0.889 (0.23) | 1.92 (0.22) | 0.875 (0.97) | 1.92 (0.60) | 0.855 (1.95) | 1.89 (1.13) |
| DPCP | 0.901 (0.43) | 1.95 (0.18) | 0.903 (0.43) | 1.96 (0.19) | 0.906 (0.48) | 1.98 (0.23) | 0.912 (0.61) | 2.01 (0.32) |
| DtACI | 0.901 (0.08) | 1.96 (0.16) | * | * | * | * | * | * |

maintain coverage around 0.9 throughout. In contrast, when $\alpha_1 = 0.9$, the smaller initial sets result in lower initial coverage. For the proposed method, coverage fluctuates around $1 - \alpha$, with slightly larger fluctuations under smaller $\epsilon$ values. For long-run coverage, smaller $\epsilon$ values initially result in lower coverage and set size due to the added noise from randomized responses. Nevertheless, all methods eventually converge to the nominal coverage level. Given the data configuration, where the coefficient vectors $\beta_t^{(0)}$ and $\beta_t^{(1)}$ initially move closer together and then diverge, the set size first increases and then decreases.

Figures 8 and 9 present the results for Case 2 and Case 3, respectively. In terms of instant coverage and long-run coverage, the patterns across Cases 1, 2, and 3 are generally consistent. Regarding set size and long-run set size, the trend in Case 2 is similar to that in Case 1, with set size initially increasing and then decreasing due to class overlap during the drift process. In contrast, Case 3 exhibits a consistently decreasing trend in set size, gradually stabilizing over time as the class distributions remain more separable, allowing the prediction sets to contract and stabilize. Figure 10 shows the results for Case 4. Under the No Drift scenario, compared with Case 1, the set sizes of all three methods decrease and their coverage remains stable overall, which is as expected. As summarized in Table 3, the results are similar to those in the regression setting: DPCP achieves higher coverage at the cost of larger set sizes. However, we emphasize that DPCP is an offline method and cannot be deployed in an online environment.

## B.3   Elec2

For the purpose of evaluating the proposed method, we utilize the Elec2 dataset, which provides a comprehensive record of electricity demand in New South Wales, Australia [Harries et al., 1999]. The dataset, compiled by the University of New South Wales, is well-suited for forecasting applications and has been widely employed in prior studies for time-series prediction tasks. We employ AR(3) to serve as the base forecasting model and aim to construct confidence intervals for the electricity demand. Rolling coverage refers to the short-term coverage calculation over a sliding window of recent data points, capturing immediate fluctuations and reflecting the model's localized response to data variability. In contrast, long-run coverage accumulates coverage over the entire data stream, providing a comprehensive view of overall performance stability.

As shown in Figure 11, rolling coverage and long-run coverage are generally stable, with coverage oscillating around the target level of $1 - \alpha$ and no noticeable drops. In the proposed method, higher privacy protection ($\epsilon = 1$) results in slightly increased fluctuations in rolling coverage, while the long-run coverage is initially lower but quickly converges to match other settings.

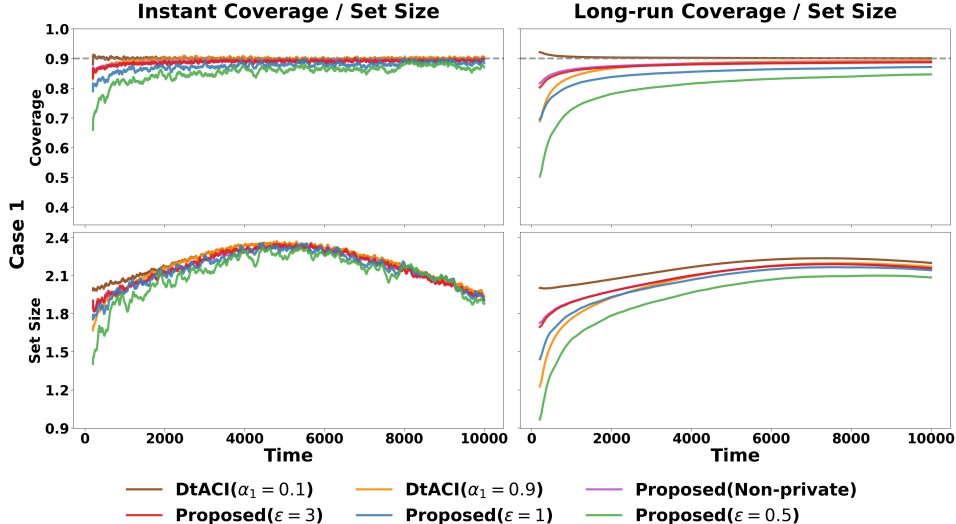

Figure 7: Simulation results for classification in Case 1, comparing different methods in terms of instant coverage, instant set size, long-run coverage, and long-run set size. Results are averaged over 200 runs; the first 200 points are excluded.

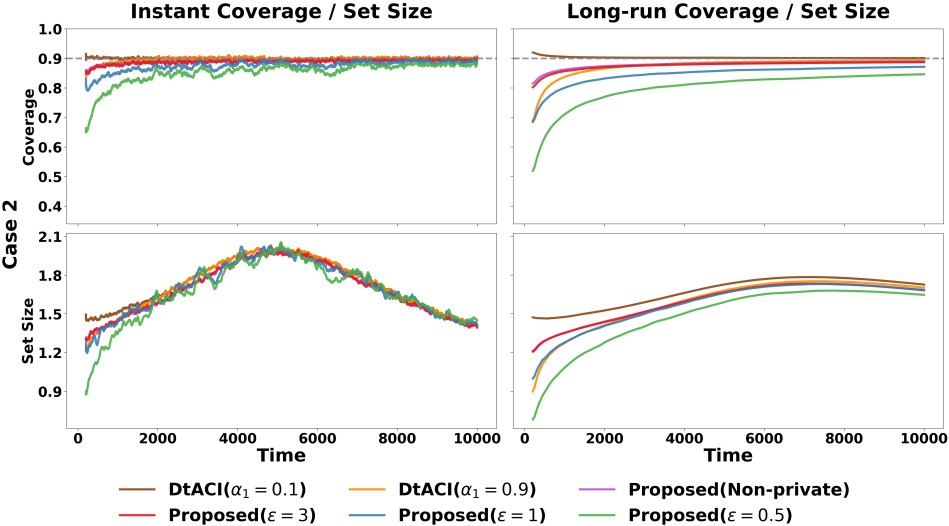

Figure 8: Simulation results for classification in Case 2, comparing different methods in terms of instant coverage, instant set size, long-run coverage, and long-run set size. Results are averaged over 200 runs; the first 200 points are excluded.

Regarding width, all methods exhibit fluctuations, capturing inherent data variability. Long-run width, however, remains relatively stable, gradually decreasing over time, suggesting that the algorithm becomes more stable as it accumulates more calibration data.

## B.4 Comparison of PTMs and Non-Private Models

In this section, we investigate the impact of model privatization on interval width and coverage by comparing privately trained models and non-private models under Case A; see Figure 12. While the long-run coverage remains unaffected, privately trained models typically exhibit lower predictive accuracy than non-private models, leading to higher non-conformity scores $S_t$ and consequently wider prediction intervals.

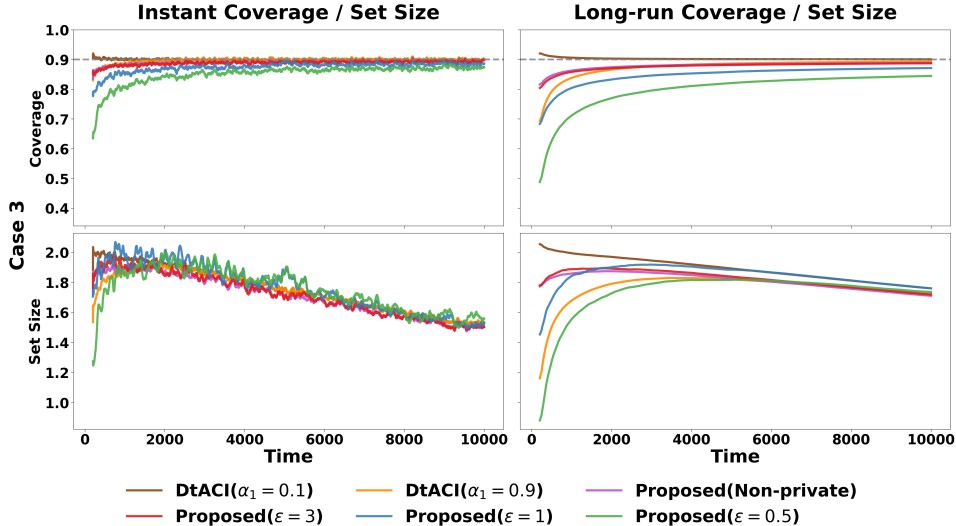

Figure 9: Simulation results for classification in Case 3, comparing different methods in terms of instant coverage, instant set size, long-run coverage, and long-run set size. Results are averaged over 200 runs; the first 200 points are excluded.

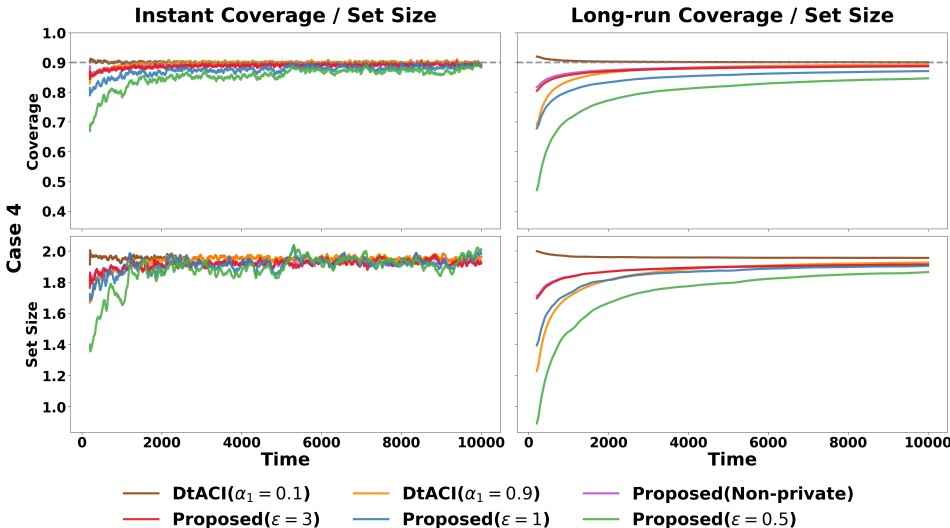

Figure 10: Simulation results for classification in Case 4, comparing different methods in terms of instant coverage, instant set size, long-run coverage, and long-run set size. Results are averaged over 200 runs; the first 200 points are excluded.

## B.5 Simulation of privacy allocation under parallel composition

We further investigate the impact of varying privacy budgets. The results in Figure 13, obtained under Case A, are consistent with the theoretical guarantees established in Theorems 3.3 and 3.4. Specifically, we observe that the performance under randomly drawn $\epsilon_t \in [0.5, 3]$ is similar to that under the fixed value $\epsilon = 3$. This is expected because the theoretical privacy bound depends on $\max_t \epsilon_t$, which, in the random case, is very likely to be close to the upper bound of 3.

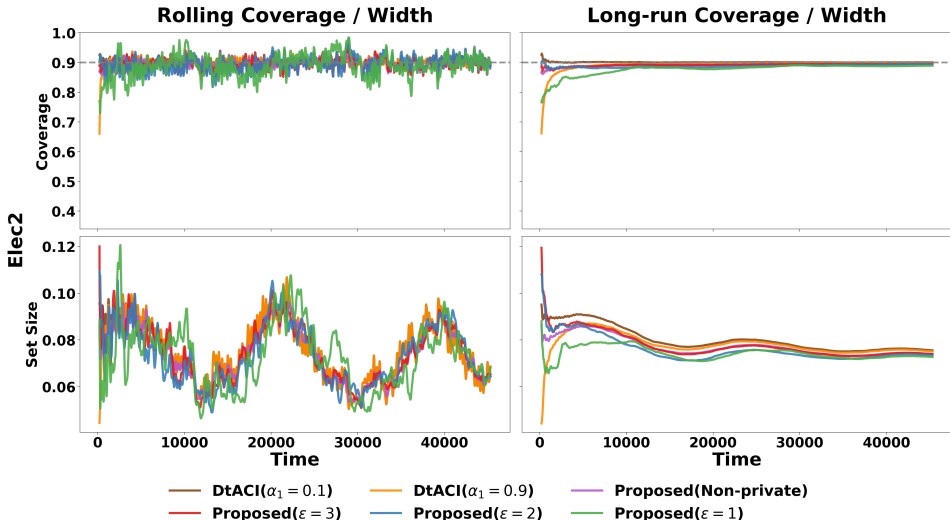

Figure 11: Results for the Elec2 dataset, comparing different methods in terms of rolling coverage/width and long-run coverage/width. The rolling metrics are computed using a sliding window of size 200, and the first 200 data points are excluded.

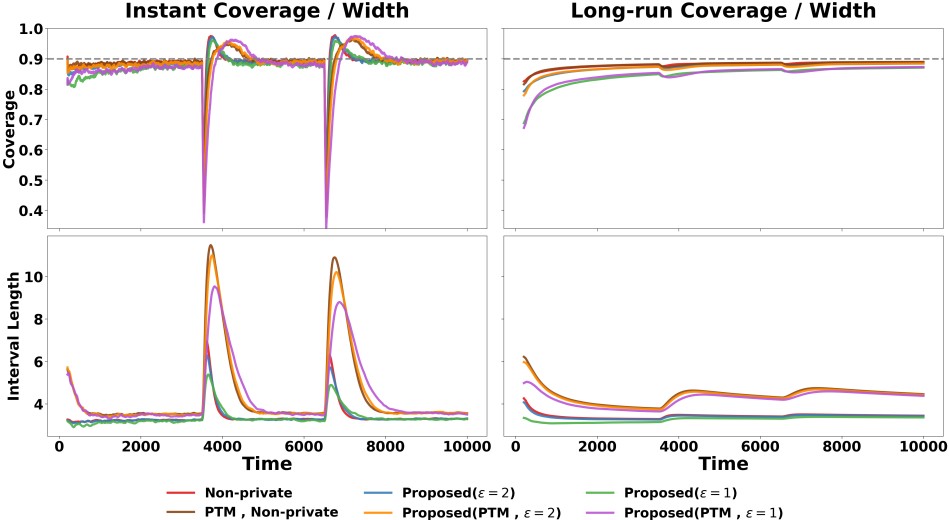

Figure 12: Comparison of instant coverage, instant interval width, long-run coverage, and long-run width between privately-trained models (PTMs) and non-private models in Case A. The results are averaged over 200 runs. The first 200 data points are excluded to mitigate initialization effects.

# C Proof

## C.1 Proof of Corollary 3.1

The stated regret bound is derived by applying the Krichevsky–Trofimov (KT) potential within the coin betting framework. In our setting, the LRBR mechanism produces privatized feedback signals $g_t$ that are uniformly bounded in magnitude, i.e., $|g_t| \leq 1$. Crucially, the coin betting framework does not require the feedback signals to be exact subgradients; it only necessitates boundedness. Therefore, the general regret bound established in Orabona and Pal [2016] remains applicable, and directly implies the result stated in Corollary 3.1.

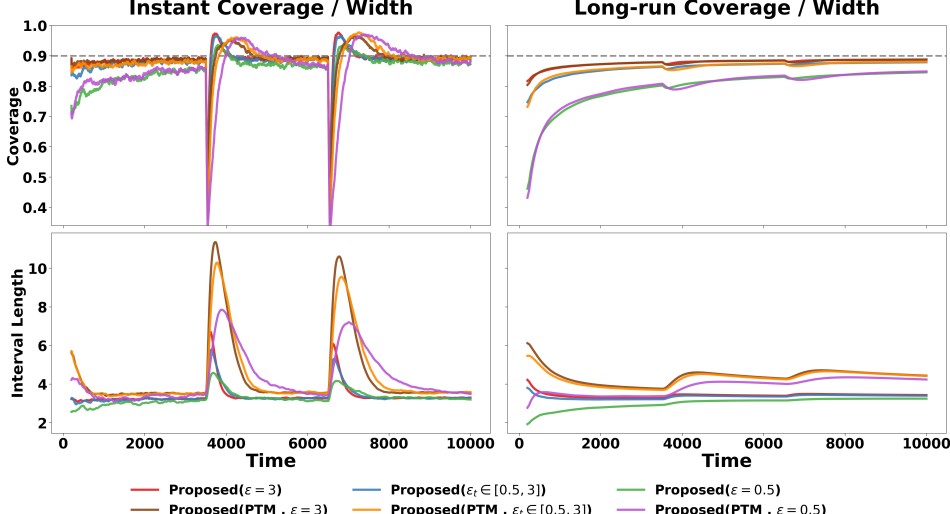

Figure 13: Simulation results in Case A, illustrating the impact of varying privacy budgets ($\epsilon_t$) under both private and non-private settings. The four panels compare instant coverage, instant width, long-run coverage, and long-run width, demonstrating how privacy parameters affect the predictive intervals over time. These results provide empirical support for the theoretical guarantees stated in Theorems 3.3 and 3.4.

## C.2 Proof of Theorem 3.2

Algorithm 1 investigates the relationship between the inquiry value $q$ and the user's binary response, controlled by the response rate $r$. Let the true label be defined as $\mathbb{I}(q > S) \in \{0, 1\}$, where $S$ is the non-conformity score. Due to local randomization, the output does not necessarily match the ground truth. The conditional probability of outputting "1" given that the truth is "1" is

$$\Pr[\text{Output} = 1 \mid \text{Truth} = 1] = \frac{1 + r}{2},$$

which occurs either when the user responds deterministically with $u = 1$ (with probability $r$), or when the individual is randomized ($u = 0$) and $v = 1$ (with probability $(1 - r)/2$). Similarly, the probability of outputting "1" given that the truth is "0" is

$$\Pr[\text{Output} = 1 \mid \text{Truth} = 0] = \frac{1 - r}{2},$$

which corresponds to the randomized case where $u = 0$ and $v = 1$.

This yields the following likelihood ratio:

$$\frac{\Pr[\text{Output} = 1 \mid \text{Truth} = 1]}{\Pr[\text{Output} = 1 \mid \text{Truth} = 0]} = \frac{(1 + r)/2}{(1 - r)/2} = \frac{1 + r}{1 - r},$$

and symmetrically,

$$\frac{\Pr[\text{Output} = 0 \mid \text{Truth} = 0]}{\Pr[\text{Output} = 0 \mid \text{Truth} = 1]} = \frac{1 + r}{1 - r}.$$

We have completed the proof of this theorem.

## C.3 Proof of Theorem 3.3

At the first iteration, Algorithm 2 is inquired via Algorithm 1 using response rate $r_1$, which induces an $(\epsilon_1, 0)$-local randomizer. By the post-processing property in Lemma 2.3, the output of the first iteration in Algorithm 2 remains $(\epsilon_1, 0)$-LDP. At the second iteration, the algorithm takes as input the privatized outputs from the first iteration, along with a new response obtained using rate $r_2$, which corresponds to an $(\epsilon_2, 0)$-local randomizer. Since the responses originate from disjoint individuals and each individual is inquired only once, the parallel composition in Lemma 2.3 ensures that the

second iteration satisfies $(\max\{\epsilon_1, \epsilon_2\}, 0)$-LDP. Using the same argument repeatedly, we conclude that after $t$ iterations, the entire Algorithm 2 satisfies $(\max_{1 \leq j \leq t} \epsilon_j, 0)$-LDP, as each iteration involves a disjoint individual and only post-processed privatized outputs are used. We have completed the proof of this theorem.

## C.4 Proof of Theorem 3.4

Notice that the first iteration is required to train a predictive model that satisfies $(\varepsilon_1, \delta_1)$-LDP, and is simultaneously inquired using Algorithm 1 with response rate $r_1$, inducing an $(\epsilon_1, 0)$-local randomizer. By the post-processing property and sequential composition stated in Lemma 2.3, the output of the first iteration in Algorithm 2 preserves $(\epsilon_1 + \varepsilon_1, \delta_1)$-LDP. Using the same arguments as in the proof of Theorem 3.3, after $t$ iterations, the entire procedure remains $(\max_{1 \leq j \leq t}(\epsilon_j + \varepsilon_j), \max_{1 \leq j \leq t} \delta_j)$-LDP via post-processing, sequential composition, and parallel composition in Lemma 2.3. The proof of this theorem is completed.

## C.5 Proof of Theorem 3.5

To simplify the notation, we define the constant $c$ as

$$c = r(1 - \alpha) + 0.5(1 - r). \tag{5}$$

**Step 1.** The feedback term $g_t$ is defined as

$$g_t = \begin{cases} 1 - c, & \text{if } L = 1 \\ -c, & \text{if } L = 0 \end{cases}.$$

The LRBR mechanism generates $L$ via the sampling steps $u \sim \text{Bernoulli}(r)$ and $v \sim \text{Bernoulli}(0.5)$. Thus, the probabilities for $L = 1$ and $L = 0$ are given by

$$\mathbb{P}(L = 1) = r \cdot \mathbb{I}\{q_t > S_t\} + (1 - r) \cdot 0.5, \quad \mathbb{P}(L = 0) = 1 - \mathbb{P}(L = 1).$$

The expectation of the feedback term is $\mathbb{E}[g_t] = (1 - c) \cdot \mathbb{P}(L = 1) + (-c) \cdot \mathbb{P}(L = 0)$, which expands to $\mathbb{E}[g_t] = r \cdot \mathbb{I}\{q_t > S_t\} + 0.5(1 - r) - c$. Substituting the definition of $c$ from (5), the expression simplifies to

$$\mathbb{E}[g_t] = r \cdot (\mathbb{I}\{q_t > S_t\} - (1 - \alpha)).$$

The cumulative sum of expectations over $T$ steps is therefore $\sum_{t=1}^{T} \mathbb{E}[g_t] = \sum_{t=1}^{T} r \cdot (\mathbb{I}\{q_t > S_t\} - (1 - \alpha))$. Rewriting the above using the coverage indicator $\mathbb{I}\{Y_t \in \hat{C}_t\} = \mathbb{I}\{q_t > S_t\}$, we obtain

$$\sum_{t=1}^{T} \mathbb{E}[g_t] = r \cdot \sum_{t=1}^{T} \left( \mathbb{I}\{Y_t \in \hat{C}_t\} - (1 - \alpha) \right).$$

Since $r > 0$ is a constant, our objective reduces to proving

$$\lim_{T \to \infty} \left| \frac{1}{T} \sum_{t=1}^{T} \mathbb{I}\left\{ Y_t \in \hat{C}_t \right\} - (1 - \alpha) \right| = 0,$$

which is equivalent to establishing

$$\lim_{T \to \infty} \left| \frac{1}{T} \sum_{t=1}^{T} \mathbb{E}[g_t] \right| = 0.$$

**Step 2.** We assume that the nonconformity scores satisfy $S_t \in [0, D]$ for all $t = 1, 2, 3, \ldots$. (a). Suppose that for some $t \geq 1$, the predicted radius $q_t$ exceeds the upper bound $D$, i.e., $q_t > D$. Given that $q_t = \lambda_t \cdot W_{t-1}$ and the wealth is nonnegative ($W_{t-1} \geq 0$), it follows that $\lambda_t > 0$. The probability that $L = 1$ is $0.5(1 + r) > 0.5$. When $L = 1$, we have $g_t = 1 - c > 0$, which implies $W_t = W_{t-1}(1 - \lambda_t g_t) < W_{t-1}$ and $\lambda_{t+1} = \frac{t}{t+1} \lambda_t - \frac{1}{t+1} g_t < \lambda_t$. Consequently, $q_{t+1} = \lambda_{t+1} W_t < q_t$, indicating a reduction in the predicted radius. The mechanism introduces

a persistent downward tendency when $q_t > D$, and the radius eventually returns to the admissible range. This prevents unbounded growth.

(b). Suppose that for some $t \geq 1$, we have $q_t \geq 0$ but $q_{t+1} < 0$. This implies that there must exist some $k$ such that $q_{t+k} > 0$. Indeed, since $q_t \geq 0$, it follows that $\lambda_t \geq 0$, while $q_{t+1} < 0$ implies $\lambda_{t+1} < 0$. Hence,

$$0 > \lambda_{t+1} = \frac{t}{t+1}\lambda_t - \frac{1}{t+1}g_t,$$

which gives $g_t > 0$. Given that $S_{t+1} \geq 0$, the probability that $L = 0$ is $0.5(1 + r) > 0.5$, and when $L = 0$, we have $g_{t+1} = -c < 0$. Consequently,

$$\lambda_{t+2} = \frac{t+1}{t+2}\lambda_{t+1} - \frac{1}{t+2}g_{t+1} = \frac{t}{t+2}\lambda_t - \frac{1}{t+2}(g_t + g_{t+1}).$$

When $\alpha < 0.5$, we have $g_t + g_{t+1} = 1 - 2c < 0$, which implies $\lambda_{t+2} > 0$. Therefore, there exists some $k \geq 2$ such that $\lambda_{t+k} > 0$, and hence $q_{t+k} > 0$. This shows that temporary excursions into the negative region are corrected by the update rule, ensuring that the radius remains nonnegative in the long run. Next, consider the update rule:

$$\begin{aligned}
q_{t+1} &= -\frac{\sum_{i=1}^{t} g_i}{t+1}\left(1 - \sum_{i=1}^{t} g_i q_i\right) \\
&= -\frac{\sum_{i=1}^{t-1} g_i}{t+1}\left(1 - \sum_{i=1}^{t-1} g_i q_i\right) - \frac{g_t}{t+1}\left(1 - \sum_{i=1}^{t-1} g_i q_i\right) + g_t q_t \cdot \frac{\sum_{i=1}^{t} g_i}{t+1} \\
&= \frac{t}{t+1}q_t + \frac{1}{t+1}\left(-g_t + g_t \sum_{i=1}^{t-1} g_i q_i + g_t q_t \sum_{i=1}^{t} g_i\right),
\end{aligned}$$

so the increment is

$$q_{t+1} - q_t = \frac{1}{t+1}\left(-q_t - g_t + g_t \sum_{i=1}^{t-1} g_i q_i + g_t q_t \sum_{i=1}^{t} g_i\right).$$

According to [Orabona and Pal, 2016, Podkopaev et al., 2024], we have the bound

$$-Dt \leq \sum_{i=1}^{t} q_i g_i \leq 1 \quad \Rightarrow \quad \left|\sum_{i=1}^{t} q_i g_i\right| \leq Dt + 1,$$

so the absolute increment satisfies:

$$|q_{t+1} - q_t| \leq \frac{1}{t+1}(D + 1 + D(t-1) + 1 + Dt) \leq 2D + 1.$$

Combining this with $q_1 = 0 \in [0, D]$ and the analysis in (a) and (b), we conclude that $|q_t| \leq 3D + 1$, i.e., the update sequence is uniformly bounded.

**Step 3.** We begin by asserting the following statement:

$$\frac{1}{T}\sum_{t=1}^{T} g_t \to 0, \quad \text{as } T \to \infty. \tag{6}$$

We proceed by contradiction. Suppose the above does not hold. That is, there exists a constant $\varepsilon > 0$ such that

$$\forall T, \exists T' > T : \frac{1}{T'}\left|\sum_{i=1}^{T'} g_i\right| \geq \varepsilon.$$

According to the KT framework, the update of $q_{t+1}$ depends on the cumulative feedback up to time $t$,

$$|q_{t+1}| = |\lambda_{t+1} W_t| = \frac{1}{t+1}\left|\sum_{i=1}^{t} g_i\right| \cdot W_t.$$

For some index $T'$, we can lower bound the update as

$$|q_{T'+1}| = \frac{1}{T'+1} \left| \sum_{i=1}^{T'} g_i \right| \cdot W_{T'} \geq \frac{T'\varepsilon}{T'+1} \cdot W_{T'}.$$

We now invoke the KT framework's lower bound on the wealth process $W_t$ [Orabona and Pal, 2016]:

$$W_t \geq \frac{1}{K\sqrt{t}} \exp\left( \frac{t}{4} \left( \frac{1}{t} \sum_{i=1}^{t} g_i \right)^2 \right).$$

Substituting into the previous inequality yields

$$|q_{T'+1}| \geq \frac{T'}{T'+1} \cdot \frac{\varepsilon}{K\sqrt{T'}} \exp\left( \frac{T'}{4} \varepsilon^2 \right).$$

This implies that for any $T$, there exists $T' > T$ such that the above inequality holds. Since the exponential term $\exp\left(\frac{T'\varepsilon^2}{4}\right)$ grows exponentially as $T' \to \infty$, it follows that the sequence $|q_{T'+1}|$ becomes unbounded as $T' \to \infty$. This contradicts the assumption that $|q_{T'+1}|$ is bounded. Thus, the initial assumption is false, and the convergence stated in (6) is established.

**Step 4.** We now prove that

$$\frac{1}{T} \sum_{t=1}^{T} \mathbb{E}[g_t] \xrightarrow[T \to \infty]{} 0$$

by invoking the Dominated Convergence Theorem (DCT). From (6), we have already shown the result

$$A_T := \frac{1}{T} \sum_{t=1}^{T} g_t \xrightarrow[T \to \infty]{} 0.$$

By construction, $g_t \in \{1 - c, -c\}$, hence

$$|g_t| \leq M := \max\{|1 - c|, |c|\} \leq 1 \implies |A_T| \leq 1 \quad \forall T \geq 1. \tag{7}$$

Choose the constant random variable $Y \equiv 1$. Since $\mathbb{E}[Y] = 1 < \infty$, and (7) gives the domination $|A_T| \leq Y$. Combining (6) and (7) with DCT yields

$$\lim_{T \to \infty} \mathbb{E}[A_T] = \mathbb{E}\left[ \lim_{T \to \infty} A_T \right] = \mathbb{E}[0] = 0.$$

Since the sum in $A_T$ is finite, expectation and summation commute:$\mathbb{E}[A_T] = \frac{1}{T} \sum_{t=1}^{T} \mathbb{E}[g_t]$. It then follows that

$$\frac{1}{T} \sum_{t=1}^{T} \mathbb{E}[g_t] \to 0, \quad \text{as } T \to \infty.$$

Since the absolute value is applied to the cumulative sum rather than each individual term, we obtain

$$\frac{1}{T} \left| \sum_{t=1}^{T} \mathbb{E}[g_t] \right| \to 0, \quad \text{as } T \to \infty.$$

Thus, the convergence of the expected feedback sequence establishes the desired convergence of the coverage error as

$$\lim_{T \to \infty} \left| \frac{1}{T} \sum_{t=1}^{T} \mathbb{I}\left\{ Y_t \in \hat{C}_t \right\} - (1 - \alpha) \right| = 0.$$

This completes the proof.

# D   Conversion table

Table 4: **Conversion table between $r$ and $\epsilon$**

| $r$ | $\epsilon$ | $r$ | $\epsilon$ |
|------|------|------|------|
| 0 | 0.00 | 0.5 | 1.10 |
| 0.05 | 0.10 | 0.55 | 1.24 |
| 0.1 | 0.20 | 0.6 | 1.39 |
| 0.15 | 0.30 | 0.65 | 1.55 |
| 0.2 | 0.40 | 0.7 | 1.73 |
| 0.25 | 0.51 | 0.75 | 1.95 |
| 0.3 | 0.62 | 0.8 | 2.20 |
| 0.35 | 0.73 | 0.85 | 2.51 |
| 0.4 | 0.85 | 0.9 | 2.94 |
| 0.45 | 0.97 | 0.95 | 3.66 |

