# OpenReview forum: "Online Locally Differentially Private Conformal Prediction via Binary Inquiries"
_NeurIPS.cc/2025/Conference — NeurIPS 2025 poster_

### Official Review · Reviewer_H3RF · 2025-06-15

**Clarity:** 3
**Significance:** 3
**Originality:** 3
**Rating:** 5
**Confidence:** 3

**Summary:**

This paper studies the problem of privacy protection of online conformal prediction or uncertainty quantification. In the continuous streaming setting, this work proposes a method to make reliable predictions while maintaining privacy of every user by using randomized response (local DP) to answer the question. The algorithm runs in one-pass and has low space complexity and high computational complexity. The method is then applied to both regression and classification tasks. The proposed method is also model agnostic. The paper also gives theoretical guarantees as well as empirical evaluations of the method to demonstrate its feasibility.

**Questions:**

I have no questions.

**Ethical Concerns:**

["NO or VERY MINOR ethics concerns only"]

**Final Justification:**

The authors addressed my concerns adequately.

**Limitations:**

The authors did address the limitations of the work in the appendix. This would be better in the main text of the paper.

**Quality:**

3

**Strengths And Weaknesses:**

Strengths:
The paper quality is pretty good, it provides a good introduction to the topic and a clear problem description. The work studies an important problem of privacy in the online setting. To my knowledge, there is no work prior to this which studies this particular setup. Privacy is an important concern in the live decision-making systems. I believe the work is significant and original, despite the simplicity of the methods used.

Weaknesses:
The proofs are all in the appendix, which is a downside of the work. I propose that some intuition of the proofs are added to the main text of the paper to have completeness. The experiments could be better explained. Also, the figures should say what the y-axis is representing.

---

> ### Author Rebuttal · Authors · 2025-07-30
>
> Thank you for your thoughtful and critical assessment. Many of your comments will help us produce a more readable and self-contained version of the paper. Below, we address each of your specific concerns in turn.
>
> - **Intuition of the Proofs**. This is an excellent comment.  We now summarize the key ideas behind our main theoretical results and will include the following in the revised version.
>
>    - **3.1. Corollary 3.1 (Regret Bound)**. The proof leverages the KT-based (Krichevsky and Trofimov, 1981) coin-betting framework, where regret remains sublinear as long as the privatized updates $g_t$ are bounded in $[-1, 1]$ (Orabona and Pál, 2016)—a condition satisfied by our randomized response mechanism.
>
>    - **3.2. Theorem 3.2 (Privacy Guarantee)**. With probability $r$, the user returns a truthful binary response indicating whether $S_t \le q_t$; otherwise, a fair coin is flipped. This randomized mechanism masks the true response, and the privacy level $\epsilon$ quantifies the worst-case distinguishability between outputs for different true answers.
>
>    - **3.3–3.4. Theorems 3.3 and 3.4 (LDP Guarantees)**. Theorem 3.3 shows our algorithm satisfies $(\max_{1 \leq j \leq t} \epsilon_j, 0)$-LDP since each user contributes only once through a local randomizer. Privacy is preserved via parallel composition, and only post-processed private outputs are used.
>
>        Theorem 3.4 extends this to the full pipeline: if the predictive model is trained under $(\varepsilon_j, \delta_j)$-LDP, then combining it with our querying mechanism yields per-round privacy of $(\epsilon_j + \varepsilon_j, \delta_j)$.
>
>   - **3.5. Theorem 3.5 (Asymptotic Coverage).** Although each feedback bit is noisy, its expectation aligns with the true subgradient:
>     $$\mathbb{E}[g_t] = r \cdot (\mathbb{I}\\{S_t \le q_t\\} - (1 - \alpha)).$$
>
>     Thus, in expectation, the updates move $q_t$ toward the target $(1 - \alpha)$ quantile. Over time, this leads to convergence of the prediction sets to the desired long-run coverage, even under strong privacy noise.
>
> - **More explanation on the experiments.**  We agree that clearer exposition would improve the interpretability of our results. While the experiments demonstrate strong performance across different privacy levels and task types (regression/classification; synthetic/real), we will revise the experimental section to enhance clarity by:
>
>   - Clarifying axis labels in all figures (e.g., y-axis denotes empirical coverage or interval width over time);
>   - Making the figure captions more descriptive;
>   - Adding brief summaries to highlight key trends observed in each figure.
>   - For example, we observe that stronger privacy (smaller $\epsilon$) leads to slightly lower empirical coverage in early rounds due to weaker update signals $g_t$, which slows the convergence of $q_t$. As shown in Figures 2 and 3, this undercoverage diminishes over time, and coverage steadily approaches the nominal level (e.g., 90%), highlighting the method’s asymptotic robustness.
>
> - **The y-axis.** We will explicitly label all y-axes in the figures. Depending on the context, the y-axis will be marked as either "Coverage" or "Prediction Interval Width/Set Size", and figure captions will be updated accordingly to improve clarity and interpretability.
>
> - **Conclusion and Limitations.** We have moved the discussion of limitations, previously placed in Appendix B due to space constraints, into the main text for completeness. The following section will appear at the end of the main body.
>   - This paper proposes a novel and practical framework for online conformal prediction under LDP. By combining randomized binary feedback with a coin-betting update scheme, the method enables one-pass, model-free prediction set construction with formal privacy guarantees. It supports both regression and classification, operates efficiently in streaming settings with constant memory. Despite its strengths, several limitations remain. First, Theorem 3.5 provides only asymptotic coverage guarantees; deriving non-asymptotic bounds under LDP is an important direction for future work. Second, we use standard nonconformity scores, which are broadly applicable but may yield conservative sets; exploring data-adaptive scores could improve efficiency.
>
> **References**
>
> Orabona, F. and Pál, D. (2016). *Coin betting and parameter-free online learning*. In _Advances in Neural Information Processing Systems_, 29.
>
> Krichevsky, R. and Trofimov, V. (1981). *The performance of universal encoding*. IEEE Trans. Inf. Theory, 27(2):199–207.

---

> > ### Comment · Reviewer_H3RF · 2025-08-04
> >
> > Thank you for your response. I think this will make the paper more contained and improve the readability.

---

> > > ### Author Response · Authors · 2025-08-04
> > >
> > > Thank you for the suggestion. We will revise the paper to improve readability in future versions and appreciate your positive feedback.

---

### Official Review · Reviewer_Cdgt · 2025-06-23

**Clarity:** 2
**Significance:** 2
**Originality:** 3
**Rating:** 4
**Confidence:** 3

**Summary:**

This paper propose an online conformal prediction procedure under local differential privacy. Theoretical guarantee and experiments results are provided to demonstrate the proformance of the proposed method.

**Questions:**

1. In line 151, interprete the statement about "This  binary structure naturally aligns with 152 a local randomization mechanism, which is crucial for satisfying LDP constraints".
2. In line 175, why can $\omega_t$ be interpreted as an estimate of the $(1 - \alpha)$-quantile? Could you provide a more intuitive explanation of lines 9–11 in Algorithm 2?
3. Is there a trade-off between privacy constraints and prediction coverage? Why does DtACI produce wider intervals compared to the proposed method and DPCP?
4. Theorem 3.5 presents an asymptotic long-run coverage guarantee. Is it possible to provide a more explicit bound to better understand the convergence rate?
5. Regarding the experimental results, the proposed method consistently exhibits undercoverage in Table 1 and Figure 2. How can this result be explained?

------------------

I have updated the score.

**Ethical Concerns:**

["NO or VERY MINOR ethics concerns only"]

**Final Justification:**

My questions have been satisfactorily addressed, and I have updated the score accordingly.

**Limitations:**

Yes, limitations are discussed in Appendix B.

**Paper Formatting Concerns:**

No.

**Quality:**

2

**Strengths And Weaknesses:**

Strengths:
- The paper proposes an intuitive way to address the  challenge of privacy-preserving uncertainty quantification in online scenario. The problem is interesting, and the motivation is well-supported through relevant literature and illustrative examples.

Weaknesses:
- The intuitive idea behind Algorithm 2 could be illustrated more clearly.

---

> ### Author Rebuttal · Authors · 2025-07-30
>
> Thank you for your thoughtful and critical assessment. Many of your comments will help us produce a more readable and self-contained version of the paper. Below, we address each of your specific concerns in turn.
>
> - **Clarification on the Intuition Behind Algorithm 2**
>
>    - Our algorithm can be viewed as a private and adaptive betting game against streaming data. Imagine a gambler estimating a target quantile (e.g., the 90th percentile) using only privatized yes/no feedback, without access to true values.
>
>   - Despite this uncertainty, the gambler adapts intelligently—tracking her "wealth," betting on the quantile's location, and adjusting based on noisy signals. This builds on the coin-betting framework of Orabona & Pál (2016), a parameter-free online learning method with regret guarantees. In their setup, a fraction $\lambda_t \in [-1, 1]$ of wealth $W_{t-1}$ is bet on an outcome $c_t \in [-1, 1]$, updating via:
>
>     $$ W_t = W\_{t-1} + \lambda_t W\_{t-1} \cdot c_t. $$
>
>     In our setting,
>        - Each round, a new nonconformity score $ S_t $ is generated.
>        - The learner compares $ S_t $ with the current quantile estimate $ q_t $ and receives privatized binary feedback via randomized response.
>        - This yields a privatized signal $ g_t $, approximating the subgradient of the pinball loss. We set $ c_t = -g_t $ to match the coin-betting formulation.
>
>   - Algorithm 2 then proceeds as follows:
>       - **Line 9:** Updates wealth $ W_t = W\_{t-1} - q_t \cdot g_t $.
>       - **Line 10:** Computes the adaptive betting fraction $ \lambda\_{t+1} $ using the KT betting strategy (Krichevsky and Trofimov, 1981), where $\lambda_{t+1} = \frac{\sum_{i=1}^{t} g_i}{t+1} = \frac{t}{t+1} \lambda_t - \frac{1}{t+1} g_t$.
>       - **Line 11:** Updates the next quantile estimate via $ q\_{t+1} = \lambda\_{t+1} W_t $.
>
>   - This yields an LDP-compliant, one-bit adaptive quantile estimator with bounded regret. The coin-betting process is detailed in lines 170–178, and we will enhance the intuition further in the revised text for clarity.
>   - Our approach follows the standard method in Orabona & Pál (2016) (Algorithm 1). Given the theoretical nature of this work, the algorithm itself may not be as intuitive, but this is an inherent aspect of its theoretical framework.
>
> - **Clarification on line 151**. The statement refers to the fact that our update signal is derived from the key term in the subgradient of the pinball loss: $ \mathbb{I}\\{S_t \le q_t\\} - (1 - \alpha), $ which depends only on whether the nonconformity score $S_t$ is less than or equal to the current estimate $q_t$. This binary structure naturally enables the use of the classical randomized response mechanism to satisfy LDP.
>    - Instead of collecting exact values, we issue a yes/no query ("Is $S_t \le q_t$?") and perturb the yes/no answer using randomized response. Since this mechanism is inherently binary, it integrates seamlessly into our algorithm without modification. The update remains efficient, private, and fully compatible with the coin-betting framework.
>    - To improve clarity, we revised line 151 as follows: "This inherent binary structure enables us to implement a local randomization mechanism via randomized response, which directly supports LDP constraints."
>
> - **Clarification on  $w_t$  and explanation of Algorithm 2 lines 9–11**. Our algorithm follows the standard coin-betting framework [Orabona & Pál, 2016], where a learner bets a fraction $\lambda_t \in [-1, 1]$ of current wealth $W_{t-1}$ on an outcome $c_t \in [-1, 1]$, yielding a signed bet $w_t = \lambda_t W_{t-1}$. This serves as a parameter-free estimate of the target quantity. In [Podkopaev et al., 2024], $c_t = -g_t$, where $g_t$ corresponds to the subgradient of the pinball loss:
>
>   - If $q_t \ne S_t$, then
>   $$
>   \partial \ell_{1-\alpha}(q_t, S_t) = \mathbb{I}\\{S_t \leq q_t\\} - (1 - \alpha)
>   $$
>
>   - If $q_t = S_t$, then
>   $$
>   \partial \ell_{1-\alpha}(q_t, S_t) \in [\alpha - 1, \alpha]
>   $$
>
>    This subgradient structure enables coin-betting to estimate the target quantile. Coin-betting implements a parameter-free      online optimization method with provable regret bounds. When applied to the pinball loss subgradient—whose expectation vanishes at the $(1 - \alpha)$-quantile—it naturally drives $w_t$ toward this quantile, yielding a consistent estimate without learning rate tuning.
>
>   - In our setting, we introduce a privatized version of $g_t$ via randomized response, carefully designed to satisfy:
>
>     $$ \mathbb{E}[g_t] = r \cdot (\mathbb{I}\\{ S_t \le q_t \\} - (1 - \alpha)), $$
>
>     which mirrors the (scaled) pinball subgradient, ensuring consistency with the key term in the loss subgradient. This preserves the alignment with quantile estimation while enabling local differential privacy. For this reason, we interpret $w_t$ as a private approximation of the $(1 - \alpha)$-quantile.
>
>   - Lines 9–11 of Algorithm 2 implement the classic coin-betting update loop, adapted to our privacy-aware setting:
>      - **Line 9** updates wealth $W_t$ based on the privatized signal $g_t$ and prediction $q_t$, reflecting gain or loss from the “bet.”
>      - **Line 10** uses the KT betting strategy to update $\lambda\_{t+1}$, acting as a running average of feedback.
>      - **Line 11** computes $q\_{t+1} = \lambda\_{t+1} W_t$, which becomes the next quantile estimate.
>
>   - This loop provides an adaptive, parameter-free quantile estimator that is simple, efficient, and LDP-compliant. We elaborate further on its coin-betting interpretation in lines 170–178 of the paper.
>
> - **On the trade-off between privacy constraints and prediction coverage**
>
>   - There is an inherent trade-off between privacy and prediction coverage: stronger privacy (i.e., smaller $\epsilon$) adds more noise, weakening the feedback signal and slowing convergence. This can lead to temporary undercoverage in early rounds, particularly in finite-sample settings (e.g., $n = 10{,}000$).
>
>    - This effect arises from the randomness introduced by the randomized response mechanism. However, as shown in Figures 2 and 3, coverage improves over time and steadily approaches the nominal level (e.g., 90%) across all privacy levels. In larger samples (e.g., Figure S.6), coverage is already close to nominal, confirming asymptotic consistency.
>
>   - In summary, while early undercoverage may occur due to privacy noise, our method achieves stable long-run coverage, demonstrating robustness even under strong LDP constraints.
>
> - **Clarification on wider intervals of DtACI.** As shown in Table 1, DtACI yields intervals that are slightly wider than ours but still much narrower than DPCP, with a slightly higher coverage. A similar pattern is seen in classification (Table S.1):
>
>   - Regarding DtACI, it produces slightly wider intervals (by ~0.03–0.04), but also provides ~0.01 higher coverage—an intuitive result of the coverage-size trade-off. The difference reflects the core design distinction: DtACI adaptively adjusts the miscoverage level $ \alpha_t $ using expert weighting, while our method directly estimates the $ (1 - \alpha) $ quantile through a parameter-free coin-betting update. Our method is feedback-efficient and tuning-free. In contrast, DtACI involves expert initialization and step size tuning, leading to more conservative calibration. These small differences reflect each method’s structure and trade-offs.
>
>
>
>
>
> **Table: Coverages and sizes for three classification cases under the proposed method and DtACI**
> |    | Case 1  | Case 1 | Case 2  | Case 2 | Case 3 | Case 3 |
> |----------|-------------|------|-------------|------|-------------|------|
> | Method   | Coverage | Size | Coverage | Size | Coverage | Size |
> | Proposed | 0.890       | 2.17 | 0.890       | 1.69 | 0.890       | 1.71 |
> | DtACI    | 0.900       | 2.20 | 0.901       | 1.73 | 0.900       | 1.75 |
>
>
>
> - **Convergence Rate**. As noted in Appendix B, deriving non-asymptotic convergence bounds under LDP is an important but challenging open problem. While our current analysis focuses on asymptotic guarantees, we plan to explore finite-sample bounds in future work. That said, empirical results already show finite-sample performance even under strong privacy budgets (e.g., $\epsilon = 0.5$), with average coverage around 0.85 over 10,000 time steps, demonstrating the method's practical robustness.
>
> - **Undercoverage in Table 1 and Figure 2:**
>
>   - We acknowledge the slight undercoverage observed in Table 1 and Figure 2. A similar trend was reported by Podkopaev et al. (2024), who observed ~88% long-run coverage using a coin-betting-based conformal predictor and considered the gap minor in practice. In our case, this undercoverage likely stems from early-round noise introduced by privatized feedback under strong privacy settings.
>
>   - This reflects a well-known trade-off: stronger privacy (smaller $\epsilon$) increases noise via randomized response, weakening update signals, slowing convergence, and causing temporary undercoverage in finite samples (e.g., $n = 10{,}000$).
>
>   - However, this effect is transient. As shown in Figures 2 and 3, coverage steadily improves and converges toward the nominal level across all privacy levels. Larger-sample results (Figure S.6) show near-complete convergence, confirming asymptotic consistency.
>
>   - In summary, while early-stage undercoverage may occur, long-run coverage remains reliable—even under strong privacy (e.g., $\epsilon = 0.5$), the method achieves 85% coverage, demonstrating practical robustness.
>
> **References**
>
> Orabona, F. and Pál, D. (2016). *Coin betting and parameter-free online learning*. In _Advances in Neural Information Processing Systems_, 29.
>
> Podkopaev, A., Xu, D., and Lee, K. (2024). *Adaptive conformal inference by betting*. arXiv preprint arXiv:2412.19318.
>
> Krichevsky, R. and Trofimov, V. (1981). *The performance of universal encoding*. IEEE Trans. Inf. Theory, 27(2):199–207.

---

> > ### Comment · Reviewer_Cdgt · 2025-08-04
> > **Acknowledgement and response to the rebuttal**
> >
> > I sincerely appreciate your efforts to carefully address my concerns! My questions have been satisfactorily addressed, and I have updated the score accordingly.

---

> > > ### Author Response · Authors · 2025-08-04
> > >
> > > We’re glad our responses addressed your concerns. Thank you for your careful reading and thoughtful feedback, which greatly improved the clarity and completeness of the paper. Please let us know if any further clarification would be helpful.

---

### Official Review · Reviewer_yJev · 2025-06-30

**Clarity:** 3
**Significance:** 2
**Originality:** 2
**Rating:** 4
**Confidence:** 2

**Summary:**

The authors introduce an online conformal prediction framework with local differential privacy in streaming data. Their method uses randomized binary queries and generates model-free prediction sets. The framework applies to regression and classification tasks with theoretical guarantees. Experimental validation on synthetic and real-world datasets confirms the method's accuracy.

**Questions:**

1. Does the framework apply only to one-dimensional problems? Equation 2 references gradients while quantiles operate in one dimension. Similarly, Algorithm 2 (line 10) describes a "gradient update" that appears to be a derivative update.

2. How do alternative methods perform under local differential privacy constraints? What are their respective performance and utility trade-offs?

3. What specific utility advantages does the proposed method offer over existing approaches?

**Ethical Concerns:**

["NO or VERY MINOR ethics concerns only"]

**Final Justification:**

The authors provide sufficient arguments in their answer, my initial concern is solved.

**Limitations:**

yes

**Paper Formatting Concerns:**

No major issues found

**Quality:**

2

**Strengths And Weaknesses:**

The paper demonstrates good writing quality and presents an intriguing local online framework for privacy-preserving settings. However, the framework appears constrained to one-dimensional learning problems due to its reliance on quantiles $(q_t)$.

The evaluation would benefit from broader comparisons with existing methods beyond the binary inquiry approach. The theoretical contribution could be strengthened by more clearly articulating the specific theoretical challenges addressed and demonstrating why this solution outperforms alternatives. Given the one-dimensional nature of the problem, gradient descent may not be the most appropriate benchmark, comparison with Newton methods would be more relevant.

---

> ### Author Rebuttal · Authors · 2025-07-29
>
> Thank you for your thoughtful and critical assessment. Many of your comments will help us produce a more readable and self-contained version of the paper. Below, we address each of your specific concerns in turn.
>
> - **Does the framework apply only to one-dimensional problems**. This is an excellent comment.
>    - Our current framework is indeed tailored for one-dimensional output tasks (e.g., scalar regression or classification), where the quantile threshold $q_t$ directly defines prediction intervals or sets.
>
>     - The main contribution lies in developing a fully online, locally differentially private CP method that requires only a single-bit privatized feedback via randomized response. To our knowledge, the first such framework for one-pass conformal prediction with binary LDP feedback.
>
>    - While multivariate outputs were not our original focus, we consider a promising way to extend to multivariate outputs. Specifically, for multivariate responses $Y_t \in \mathbb{R}^p$, we define a scalar nonconformity score using an ellipsoidal projection:
>      $$ \hat{e}(Y) = (Y - \hat{f}(X_t) - \bar{\varepsilon})^T \hat{\Sigma}^{-1} (Y - \hat{f}(X_t) - \bar{\varepsilon}),$$
>      where $\hat{\Sigma} $ is a low-rank approximation of the residual covariance matrix based on residuals
>     $ \hat{\varepsilon}_t = Y_t - \hat{f}(X_t) \in \mathbb{R}^p.$
>    This score can be used in our privacy-preserving framework to learn the privacy quantile $\hat{Q}_t(1 - \alpha)$, forming an ellipsoidal prediction region:
>
>      $$ \hat{C}_{t-1}(X_t, \alpha) = \left\\{ Y : \hat{e}(Y) \leq \hat{Q}_t(1 - \alpha) \right\\}. $$
>
>    - This extension is aligned with prior work (e.g. Xu et al., 2024 and Braun et al., 2025) who use Mahalanobis-type distances for multivariate conformity, demonstrating the flexibility of our framework beyond the univariate case.
>
> - **Comparison with Newton methods**
>
>    - Classic online cp methods (e.g., Angelopoulos et al., 2023; Podkopaev et al., 2024) often formulate quantile estimation as online optimization of the pinball loss:
>
>      $$\ell\_{1-\alpha}(q, S_t) = (\mathbb{I}\\{q \geq S_t\\} - (1 - \alpha))(q - S_t),$$
>
>      which is convex but non-smooth and admits only subgradients.
>
>    - Our approach adopts a coin-betting strategy that is parameter-free, step size-free, and single-pass, making it well-suited for streaming and LDP settings. Unlike gradient-based methods, it does not rely on smoothness or differentiability.
>
>   - In contrast, Newton methods require twice-differentiable loss functions. Since the pinball loss is non-differentiable at $q = S_t$, Newton-style updates are inapplicable in this context.
>
> - **Clarification on Equation 2 and the "gradient update" in Algorithm 2**. We have revised the terminology for clarity and consistency with standard conventions in online convex optimization for non-smooth problems:
>   - In Algorithm 2 (line 10), we now refer to the update simply as an update, rather than a “gradient update.”
>   - In Equation (2), we clarify that the update is based on a subgradient of the pinball loss.
>   - The symbol has been corrected from $\nabla \ell_{1-\alpha}$ to $\partial \ell_{1-\alpha}$ to accurately reflect the use of a subgradient (not a gradient), given the non-differentiability of the pinball loss at $q = S_t$.
>
> - **Broader comparison and justification of the update mechanism**. Existing privacy-preserving CP methods are limited in number and mostly focus on the offline setting. We compare our method against two recent and representative approaches:
>    - Penso et al. (2025) proposed two offline methods: one perturbs labels via $k$-ary randomized response, the other perturbs conformity scores using binary randomized queries. Both achieve finite-sample coverage without ground-truth labels but require disjoint calibration batches and fixed datasets, limiting applicability in streaming settings. In contrast, our method supports fully online prediction with $O(1)$ memory and single-pass quantile updates. Moreover, our model-agnostic framework accommodates both regression and classification, unlike their classification-specific approach dependent on the label space size $k$.
>
>   - Zhang et al. (2025) introduced ODPCP, which adds noise to subgradients and applies truncation. This method requires hyperparameter tuning and sensitivity calibration. By contrast, our approach employs parameter-free binary randomized response (aside from $\epsilon$), integrates seamlessly with coin betting, and enjoys bounded regret guarantees. Prior work (e.g., Balle et al., 2018; Liu et al., 2022) shows that randomized response more efficiently utilizes the privacy budget compared to additive noise.
>
>    - We also conduct simulation studies comparing our method with ODPCP across various privacy levels. As shown, our method yields tighter and more stable intervals than ODPCP, consistent with the instability reported in Zhang et al. (2025, Fig. 15) for low privacy budgets.
>
> **Table: Proposed method vs ODPCP**
> (Std devs: coverage ×10⁻², width ×10⁻¹)
> |  |   | $\epsilon=3$ | $\epsilon=3$   | $\epsilon=1$ | $\epsilon=1$         | $\epsilon=0.5$  | $\epsilon=0.5$    |
> |------|----------|------------------------|---------------|------------------------|---------------|--------------------------|---------------|
> | Case | Method   |  Cov. | Width | Cov. | Width  |  Cov. | Width   |
> | A    | Proposed | 0.889 (0.2)            | 3.25 (0.4)    | 0.875 (1.0)            | 3.16 (1.1)    | 0.853 (2.0)              | 3.11 (2.5)    |
> |    | ODPCP    | 0.888 (0.5)            | 3.26 (0.6)    | 0.868 (1.4)            | 3.94 (10.0)   | 0.864 (2.6)              | 7.42 (54.0)   |
> | B    | Proposed | 0.889 (0.3)            | 4.16 (1.2)    | 0.874 (0.9)            | 3.94 (2.5)    | 0.851 (1.8)              | 3.60 (4.3)    |
> |     | ODPCP    | 0.888 (0.5)            | 4.20 (1.6)    | 0.869 (1.3)            | 4.72 (11.0)   | 0.865 (2.4)              | 8.59 (54.0)   |
> | C    | Proposed | 0.889 (0.3)            | 3.10 (0.4)    | 0.875 (1.1)            | 3.04 (1.0)    | 0.852 (1.9)              | 2.95 (1.8)    |
> |     | ODPCP    | 0.888 (0.5)            | 3.13 (0.6)    | 0.869 (1.4)            | 3.61 (23.0)   | 0.867 (2.7)              | 6.65 (30.0)   |
>
> - **Theoretical Contributions.** We highlight the following theoretical contributions:
>    - Our method inherits regret guarantees from the coin-betting framework by using a privacy-aware update rule $g_t \in [-1,1]$, enabled by binary randomized response. In contrast, additive-noise methods (e.g., Zhang et al., 2025) cannot ensure bounded updates and thus lack similar guarantees.
>   - We establish formal LDP guarantees in the fully online setting, addressing a gap in the literature where most theoretical results focus on offline calibration.
>   - The update rule satisfies: $\mathbb{E}[g_t] = r \cdot (\mathbb{I}\\{ S_t \le q_t \\} - (1 - \alpha)),$ which mirrors the subgradient of the pinball loss. This structure allows us to prove asymptotic coverage guarantees despite the presence of privacy noise.
>
> - **Utility trade-offs and coverage behavior.** Privacy constraints inherently affect coverage performance. In offline methods (e.g., Angelopoulos et al., 2022; Penso et al., 2025), stronger privacy (smaller $\epsilon$) typically leads to over-coverage and wider intervals due to conservative calibration. For example, Theorem 2 in Angelopoulos et al. (2022) shows that:
>
>   $$ \tilde{q} = \frac{(n+1)(1-\alpha)}{n(1-\gamma \alpha)} + \frac{2}{\epsilon n} \log \left( \frac{m}{\gamma \alpha} \right), $$
>
>   where the second term increases as $\epsilon$ decreases, inflating $\tilde{q}$ and widening the prediction interval.
>
>    In contrast, online methods (e.g., Zhang et al., 2025; ours) tend to exhibit under-coverage, especially in early rounds. This is because privacy noise, additive in ODPCP or randomized in our approach, dampens the update signal, slowing the convergence of quantile estimates.
>
> - **Advantages of the proposed method.** Our method offers several key advantages over existing LDP-based conformal prediction approaches:
>   - Efficiency: It supports single-pass updates with constant memory, unlike offline methods (e.g., Angelopoulos et al., 2022; Penso et al., 2025) that require fixed calibration sets and multiple passes, making it well-suited for streaming settings.
>
>   - Parameter-free: In contrast to additive-noise methods (e.g., Zhang et al., 2025) that require sensitivity tuning, our method only depends on the privacy budget $\epsilon$, simplifying implementation and improving robustness.
>
>   - Empirical strength: As shown in Table 1 and Figure 2, our method achieves strong coverage with tighter intervals even under strong privacy $\epsilon = 0.5$, balancing privacy and utility.
>
>   - Generality: Unlike classification-specific approaches (e.g., Penso et al., 2025), our framework is model-agnostic and applicable to both regression and classification tasks.
>
> **References**
>
> Angelopoulos et al., 2023. *Conformal PID control for time series prediction*. _NeurIPS_.
>
> Angelopoulos et al., 2022. *Private prediction sets*. _Harvard Data Science Review_.
>
> Balle et al., 2018. *Privacy amplification by subsampling: Tight analyses via couplings and divergences*. _NeurIPS_.
>
> Braun et al., 2025. *Multivariate conformal prediction via conformalized Gaussian scoring*. arXiv.
>
> Liu et al., 2022. *Identification, amplification and measurement: A bridge to Gaussian differential privacy*. _NeurIPS_.
>
> Orabona and Pál, 2016. *Coin betting and parameter-free online learning*. _NeurIPS_.
>
> Penso et al., 2025. *Privacy-preserving conformal prediction under local differential privacy*. arXiv.
>
> Podkopaev et al., 2024. *Adaptive conformal inference by betting*. arXiv.
>
> Xu et al., 2024. *Conformal prediction for multi-dimensional time series by ellipsoidal sets*. arXiv.
>
> Zhang et al., 2025. *Online differentially private conformal prediction for uncertainty quantification*. _ICML_.

---

> ### Author Response · Authors · 2025-08-07
>
> Dear Reviewer yJev,
>
> Thank you once again for your thoughtful and constructive assessment, and for the time and effort you have dedicated to reviewing our submission.
> We would greatly appreciate it if you could let us know whether our responses above have adequately addressed your concerns.
> If you have any additional comments or questions, we would be grateful for the opportunity to further clarify.
>
> Sincerely,
>
> The Authors

---

> ### Comment · Reviewer_yJev · 2025-08-07
>
> I would like to thank the authors for the detailed answers and putting great effort to compare with different results. My initial concerns are solved, even thought I still wonder how would the lower bound be for this class of problem. I don't see where to change the score but consider all other answers that the authors provide during the rebuttal phase, I would increased the score up to 4.

---

> > ### Author Response · Authors · 2025-08-08
> >
> > We sincerely appreciate your insightful follow-up feedback and your willingness to reassess our work. In response to your additional questions, we provide the following clarifications:
> >
> > - **Lower bound for this class of problem**. This is an excellent comment. We noticed that regret lower bound analysis remains largely unexplored in the privacy-preserving and online conformal prediction literature.
> >
> >    - **Privacy-preserving conformal prediction.** Angelopoulos et al. (2022) were among the first to propose a privacy-preserving method, but their approach is limited to the offline setting, where regret analysis is not applicable. Subsequent works by Plassier et al. (2023), Humbert et al. (2023), Penso et al. (2025) and Romanus & Molinari (2025) also focused on offline scenarios, where regret is not well-defined due to the lack of online feedback and iterative updates. A concurrent work by Zhang et al. (2025) considers the online setting and provides long-term coverage guarantees but does not establish formal regret bounds. Therefore, regret analysis remains largely unexplored in the context of privacy-preserving conformal prediction.
> >
> >    - **Non-private online conformal prediction.** Regret lower bounds have been rarely studied in the context of online conformal prediction. For example, prior works such as Gibbs et al. (2021), Zaffran et al. (2022), Gibbs & Candès (2024), Angelopoulos et al. (2024) and Podkopaev et al. (2024) do not mention regret lower bounds. Instead, these studies primarily focus on coverage guarantees, with some also providing regret upper bounds. Gibbs et al. (2021), for instance, were among the first to introduce adaptive conformal prediction, derived  uniform deviation bounds (Proposition 4.1 in Gibbs et al., 2021) to ensure empirical error control instead of analyzing regret directly.
> >
> >
> >   Due to the added complexity of incorporating privacy mechanisms (e.g., binary randomized response) with coin betting and conformal prediction, we do not attempt to derive regret lower bounds in this work. We view this as a foundational and technically challenging direction for future research. One promising avenue is to leverage information-theoretic tools (e.g., Fano’s inequality or mutual information bounds) to characterize the privacy–utility trade-off in terms of both miscoverage and lower bounds on cumulative regret. We leave a formal lower-bound analysis to future work and will include this discussion in the *Conclusion and Discussion* Section.
> >
> > While we do not analyze regret lower bounds in this work, our theoretical contributions focus on privacy guarantees (Theorems 3.2–3.4) and coverage validity (Theorem 3.5).
> >
> > By the way, you may use the *Edit* button at the top right of your initial review comment to update the score and submit your final rating.
> >
> > Thank you again for your thoughtful feedback and for raising such a meaningful and technically insightful question.
> >
> >
> > ### References
> >
> > Angelopoulos et al. (2022). *Private prediction sets*. _Harvard Data Sci. Rev_.
> >
> > Plassier et al. (2023). *Conformal prediction for federated uncertainty quantification under label shift*. _ICML 2023_.
> >
> > Humbert et al. (2023). *One-shot federated conformal prediction*. _ICML 2023_.
> >
> > Penso et al. (2025). *Privacy-Preserving Conformal Prediction Under Local Differential Privacy*. _arXiv:2505.15721_.
> >
> > Romanus & Molinari (2025). *Differentially Private Conformal Prediction via Quantile Binary Search*. _arXiv:2507.12497_.
> >
> > Zhang et al. (2025). *Online Differentially Private Conformal Prediction for Uncertainty Quantification*. _ICML 2025_.
> >
> > Gibbs et al. (2021). *Adaptive Conformal Inference Under Distribution Shift*. _NeurIPS 2021_.
> >
> > Zaffran et al. (2022). *Adaptive Conformal Predictions for Time Series*. _ICML 2022_.
> >
> > Gibbs & Candès (2024). *Conformal Inference for Online Prediction with Arbitrary Distribution Shifts*. _JMLR_.
> >
> > Podkopaev et al. (2024). *Adaptive Conformal Inference by Betting*. _arXiv_.
> >
> > Angelopoulos et al. (2024), *Online conformal prediction with decaying step sizes*, _ICML 2024_.

---

### Official Review · Reviewer_RLiw · 2025-06-30

**Clarity:** 4
**Significance:** 3
**Originality:** 3
**Rating:** 5
**Confidence:** 3

**Summary:**

This paper introduces an online conformal prediction (CP) method that produces locally differentially private quantiles. Based on randomized binary inquiries, the method is computationally efficient and enables to dynamically construct distribution-free prediction sets that maintain privacy. Furthermore, the authors demonstrate that the long-run empirical coverage (eq. 1) converges to a pre-specified nominal level $1−\alpha$ and that the cumulative regret is controlled (Corollary 3.1). Finally, on several experiments, they illustrate the efficiency of their method.

**Questions:**

1\ In Algorithm 2, the value of $r_t$​ depends on time. What is a practical application where we would want to modify the degree of privacy over time?

2\ What is the purpose of Section 4, "Extension to Classification Tasks"? It seems to be just a modification of the score function.

3\ In the experiments, Table 1 shows that coverages of the sets returned by the proposed method are lower than 0.9, even if the number of points is huge (10,000). Is there an explanation for this?

4\ Furthermore, in Table 1, we see that when $\epsilon$ increases, the coverage decreases for the proposed method, which is the opposite of what is observed for DPCP and seems more natural. Can you elaborate on this?

5\ Do you think it could be possible to see the impact of the privacy in the regret bound in Corollary 3.1?

6\ (Minor) When there is no privacy, is DPCP exactly equal to the standard split method?

**Ethical Concerns:**

["NO or VERY MINOR ethics concerns only"]

**Final Justification:**

In my opinion, this is a good article. After the rebuttal I keep my score.

**Limitations:**

yes

**Paper Formatting Concerns:**

everything seems ok except that there is no section "Conclusion"

**Quality:**

3

**Strengths And Weaknesses:**

### Strengths

1\ The paper is very well-written and easy to follow.  Furthermore, the problem is well-motivated.

2\ The method is well-explained, clever, and simple in a positive manner.

3\ The subject is of particular interest to the CP community.

###  Weaknesses

Overall, to me, there are not many drawbacks to this paper. However, here are a few:

1\ The bound on the regret in Corollary 3.1 does not depend on $r_t$ (the privacy).

2\ The tables in the experiments do not include standard deviations.

3\ In line 271, it is stated that "The proposed method achieves slightly lower but comparable coverage to DPCP, while producing substantially narrower intervals." This is an interesting remark, but to properly compare the sets, they should be compared when they have the same size or coverage.

4\ There is no "Conclusion and Limitations" section.

Typos: Line 98: "of of"

---

> ### Author Rebuttal · Authors · 2025-07-30
>
> Thank you for your thoughtful and critical assessment. Many of your comments will help us produce a more readable and self-contained version of the paper. Below, we address each of your specific concerns in turn.
>
> - **Standard deviations in the experiment.**  We have added a table reporting the standard deviations of coverage and prediction interval width, which will be included in the revised version.
> > **Table: Standard deviations of coverage (×10⁻²) and prediction interval width (×10⁻¹).**
>
> |       |         | no-privacy | no-privacy |   ε = 3   |  ε = 3    |   ε = 1   |  ε = 1    | ε = 0.5  |  ε = 0.5   |
> |-------|---------|------------|------------|-----------|-----------|-----------|-----------|----------|-----------|
> | Case  | Method  | Coverage   | Width      | Coverage  | Width     | Coverage  | Width     | Coverage | Width     |
> | A     | Proposed| 0.890 (0.03) | 3.25 (0.3) | 0.889 (0.2) | 3.25 (0.4) | 0.875 (1.0) | 3.16 (1.1) | 0.853 (2.0) | 3.11 (2.5) |
> |       | DPCP    | 0.900 (0.5)  | 8.30 (1.0) | 0.904 (0.5) | 8.41 (1.1) | 0.911 (0.6) | 8.61 (1.5) | 0.922 (0.9) | 8.95 (3.0) |
> |       | DtACI   | 0.897 (0.08) | 3.42 (0.3) | *         | *         | *         | *         | *        | *         |
> | B     | Proposed| 0.890 (0.04) | 4.18 (0.9) | 0.889 (0.3) | 4.16 (1.2) | 0.874 (0.9) | 3.94 (2.5) | 0.851 (1.8) | 3.60 (4.3) |
> |       | DPCP    | 0.899 (0.5)  | 8.99 (1.4) | 0.903 (0.5) | 9.12 (1.6) | 0.910 (0.6) | 9.37 (2.2) | 0.921 (0.9) | 9.81 (3.9) |
> |       | DtACI   | 0.899 (0.07) | 4.82 (1.0) | *         | *         | *         | *         | *        | *         |
> | C     | Proposed| 0.890 (0.03) | 3.10 (0.3) | 0.889 (0.3) | 3.10 (0.4) | 0.875 (1.1) | 3.04 (1.0) | 0.852 (1.9) | 2.95 (1.8) |
> |       | DPCP    | 0.900 (0.5)  | 4.43 (0.6) | 0.904 (0.5) | 4.50 (0.6) | 0.912 (0.6) | 4.61 (0.9) | 0.923 (0.9) | 4.81 (1.8) |
> |       | DtACI   | 0.899 (0.08) | 3.34 (0.3) | *         | *         | *         | *         | *        | *         |
>
> - **Comparisons with the same size or coverage.**
>   - In our experiments, all methods are evaluated under a fixed nominal level $ \alpha = 0.1 $ to reflect real-world deployment scenarios.
>   - We agree that aligning empirical coverage offers a fairer comparison. Following your suggestion, we adjusted the $\alpha$ for our method to match DPCP’s empirical coverage under each privacy level. The resulting interval widths are reported below.
>   - The results show that our method produces narrower intervals while matching the empirical coverage of DPCP.
>
> > **Table: Coverages and widths for the proposed method and DPCP (coverage-matched comparison, coverage Std × 10⁻², width Std × 10⁻¹).**
> |       |         |   ε = 3   |  ε = 3    |   ε = 1   |  ε = 1    | ε = 0.5  |  ε = 0.5   |
> |-------|---------|-----------|-----------|-----------|-----------|----------|-----------|
> | Case  | Method  | Coverage  | Width     | Coverage  | Width     | Coverage | Width     |
> | A     | Proposed | 0.904 (0.2) | 3.44 (0.4) | 0.911 (1.0) | 3.69 (1.9) | 0.922 (1.9) | 4.54 (13) |
> |      | DPCP     |  0.904 (0.5) | 8.41 (1.1) | 0.911 (0.6) | 8.61 (1.5) | 0.922 (0.9) | 8.95 (3.0) |
> | B     | Proposed |  0.903 (0.3) | 4.68 (1.4) | 0.910 (0.9) | 5.34 (4.5) | 0.921 (1.8) | 7.08 (18) |
> |      | DPCP     |  0.903 (0.5) | 9.12 (1.6) | 0.910 (0.6) | 9.37 (2.2) | 0.921 (0.9) | 9.81 (3.9) |
> | C     | Proposed |  0.904 (0.3) | 3.28 (0.4) | 0.912 (1.1) | 3.51 (1.6) | 0.923 (1.8) | 4.10 (6.4) |
> |      | DPCP     |  0.904 (0.5) | 4.50 (0.6) | 0.912 (0.6) | 4.61 (0.9) | 0.923 (0.9) | 4.81 (1.8) |
>
> - **Conclusion section.** In the revised version, we will include the following section including conclusions and limitations of this paper.
>    - This paper proposes a novel and practical framework for online conformal prediction under LDP. By combining randomized binary feedback with a coin-betting update scheme, the method enables one-pass, model-free prediction set construction with formal privacy guarantees. It supports both regression and classification, operates efficiently in streaming settings with constant memory. Despite its strengths, several limitations remain. First, Theorem 3.5 provides only asymptotic coverage guarantees; deriving non-asymptotic bounds under LDP is an important direction for future work. Second, we use standard nonconformity scores, which are broadly applicable but may yield conservative sets; exploring data-adaptive scores could improve efficiency.
>
>
> - **Clarification on $r_t$**. The mechanism is not inherently time-dependent; rather, each time step typically represents a new user in the online setting, with $r_t$ reflecting that user's individual privacy preference. In practice, users may differ in their desired level of privacy—privacy-conscious users may choose a smaller $r_t$, introducing more noise, while others may opt for a larger $r_t$ to allow more informative updates. This flexibility makes the algorithm suitable for real-world scenarios with heterogeneous privacy preferences.
>
> - **Purpose of Section 4**. While Section 4 changes the nonconformity score, this is not a mere technical detail—it illustrates the framework’s applicability to both regression and classification. Unlike contemporaneous work (e.g., Penso et al., 2025), which focuses solely on classification, our method is model- and task-agnostic, applicable to any setting with a scalar nonconformity score. Section 4 demonstrates how the same core algorithm (Algorithm 2) extends to classification with minimal adjustments, reinforcing our claim of task generality and supporting clarity and reproducibility.
>
>
> - **Coverages are lower than 0.9**. This behavior is consistent with prior findings. Podkopaev et al. (2024) reported similar long-run coverage (~88\%) for their coin-betting conformal predictor even without privacy, and considered the small gap a minor practical issue. In our case, stronger privacy (i.e., smaller $\epsilon$) introduces more randomness via randomized response, which weakens the update signal and slows convergence—especially in early rounds—leading to temporary undercoverage in finite samples. Nonetheless, as shown in Figures 2, 3, and S.6, coverage steadily improves with more data and approaches the nominal 90\%, confirming asymptotic consistency. Even under strong privacy (e.g., $\epsilon = 0.5$), the method maintains ~85\% coverage, demonstrating solid practical performance.
>
> - **Why does our coverage drop as $ \epsilon $ decreases (opposite DPCP, more intuitive)**. This is an excellent comment.
>   - DPCP is an offline method that privatizes calibration scores using the exponential mechanism (Angelopoulos et al., 2022). Specifically, their Equation (3):
>   $$
> \tilde{q} = \frac{(n+1)(1-\alpha)}{n(1-\gamma\alpha)} + \frac{2}{\epsilon n}\log\left(\frac{m}{\gamma\alpha}\right)
>  $$
>     shows that the privatized quantile $\tilde{q}$ increases as $\epsilon$ decreases. This inflates prediction thresholds, yielding wider, more conservative sets to preserve coverage—explaining DPCP's increasing coverage under stronger privacy.
>
>   - In contrast, our fully online method uses privatized binary feedback to update the quantile estimate. Lower $\epsilon$ reduces the response rate (e.g., $\epsilon = 0.5$ when  $r = 0.25$), increasing noise and slowing learning. This leads to slight undercoverage early on, but Figures 2 and 3 show that coverage stabilizes over time, especially after initial rounds—common in streaming scenarios.
>
>   - In summary, the coverage gap reflects early-stage randomness under privacy constraints and does not undermine the method’s strong long-term performance.
>
> - **Regret bound with respect to $r_t$ in Corollary 3.1**
>   - The regret bound in Corollary 3.1 is directly derived from the coin-betting framework of Orabona & Pál (2016) (see Page 7, Corollary 5), which guarantees bounded regret as long as the feedback signal $g_t \in [-1, 1]$ remains bounded.
>
>    - From a theoretical standpoint, the influence of the privacy parameter $r$ is absorbed into the worst-case analysis inherent to coin-betting. To explicitly reflect the effect of $r$ in the regret bound would require substantial modifications to the original analysis (see Appendix F, Pages 19–22), which is nontrivial and beyond the current scope. Nevertheless, we agree that this is an interesting direction for future theoretical work.
>
> - **When there is no privacy, is DPCP exactly equal to the standard split method?**. You are right.  As the privacy budget $\epsilon \to \infty$ and $\gamma \to 0$, the privatized quantile $\tilde{q}$ in Equation (3) of Angelopoulos et al. (2022) converges to the standard split conformal quantile. This is explicitly noted in their paper, confirming that DPCP reduces to the standard split conformal method when privacy constraints are removed. Since DPCP builds on the split method with an added privacy mechanism, the two are equivalent in the non-private setting.
>
> - **Typos and Formatting**. We will correct the typo in Line 98 (“of of”) and perform a thorough pass over the manuscript to ensure all grammatical and formatting issues are addressed in the final version.
>
> **References**
>
> Angelopoulos, A., Bates, S., Zrnic, T., & Jordan, M. I. (2022). *Private prediction sets*. Harvard Data Science Review, 4(2).
>
> Podkopaev, A., Xu, D., & Lee, K. (2024). *Adaptive conformal inference by betting*. arXiv preprint arXiv:2412.19318.
>
> Orabona, F. and Pál, D. (2016). *Coin betting and parameter-free online learning*. In _Advances in Neural Information Processing Systems_, 29.
>
> Penso, C., Mahpud, B., Goldberger, J., and Sheffet, O. (2025). *Privacy-preserving conformal prediction under local differential privacy*. arXiv preprint arXiv:2505.15721.

---

> ### Comment · Reviewer_RLiw · 2025-08-05
>
> Thank you for the detailed response. I remain positive about the paper.

---

> > ### Author Response · Authors · 2025-08-05
> >
> > Thank you for your positive assessment of our work. We will carefully incorporate your suggestions to improve the final version of the manuscript.

---

### Official Review · Reviewer_nEbK · 2025-07-01

**Clarity:** 4
**Significance:** 3
**Originality:** 2
**Rating:** 4
**Confidence:** 3

**Summary:**

The paper studies the problem of local-DP (LDP) streaming bit information for conformal prediction (getting value within an interval with % probability guarantees). The paper:
- propose the new algorithm using the known coin betting technique + randomized coin (for DP)
- show the regret and experiments for this new algorithm

**Questions:**

1. DPCP is offline / central DP method. Will this be an apple-to-apple comparison? I feel a setting where distribution is not shifted (i.e. learning the confidence quickly) should be added because it's offline?
2. In the baseline for non-private you used DtACI. Can you explain why not Podkopaev et al?
3. Can you comment how your paper compares to Penso et al. 2025 – “Privacy-Preserving Conformal Prediction under Local DP.” and Zhang et al. 2025 – “Online Differentially Private Conformal Prediction for Uncertainty Quantification.” ?

**Ethical Concerns:**

["NO or VERY MINOR ethics concerns only"]

**Final Justification:**

Overall I see slight weakening of acceptance due to worse performance with distribution doesn't shift, as I suspected. However, there are still theoretical and ideation contributions that I think the paper can still be accepted.

The comment on multi dimensional by Reviewer yJev is also not a big concern to me.

**Quality:**

3

**Strengths And Weaknesses:**

(+) new settings of online (one-pass) + local DP + one-bit
(+) complete theory
(+) many datasets tried
(-) Analysis is mostly off-the-shelf: applying the same known technique to a slightly different algorithm. The novelty is more on settings it applies to rather than advancing any theory or new ideas to work elsewhere.
(-) Unclear / possibly missing comparison in experiment results I want to ask, see below.

---

> ### Author Rebuttal · Authors · 2025-07-29
>
> Thank you for your thoughtful and critical assessment. Many of your comments will help us produce a more readable and self-contained version of the paper. Below, we address each of your specific concerns in turn.
>
> - **Novelty, Originality, and Theoretical Contributions.** Classical online CP methods (e.g., Gibbs & Candès, 2021; Angelopoulos et al., 2023; Gibbs & Candès, 2024; Podkopaev et al., 2024) rely on true label feedback to adjust prediction intervals via pinball loss minimization: $q_{t+1} = q_t - \eta_t \cdot g_t,$ where $g_t \in \partial \ell_{1-\alpha}(q_t, S_t)$ is a subgradient of the loss.
>
>   - The subgradient is given by:
>
>     - If $q_t \ne S_t$, then
>       $$\partial \ell_{1-\alpha}(q_t, S_t) = \mathbb{I}\\{S_t \leq q_t\\} - (1 - \alpha)$$
>     - If $q_t = S_t$, then
>       $$\partial \ell_{1-\alpha}(q_t, S_t) \in [\alpha - 1, \alpha]$$
>
>   - In contrast, we propose a novel approach that replaces true feedback with randomized response, enabling privacy-preserving one-bit feedback without revealing true inclusion status—an idea not explored in prior online CP work. Our privacy-aware update mimics subgradient behavior:
>
>     - If $L = 1$,
>       $$ g_t = 1 - \left( r (1 - \alpha) + 0.5 (1 - r) \right). $$
>     - Otherwise,
>       $$ g_t = - \left( r (1 - \alpha) + 0.5 (1 - r) \right). $$
>
>   - This update ensures convergence to the target quantile while preserving user privacy. While inspired by coin-betting methods, our consideration randomized response mechanism is novel, offering a simple, efficient, and privacy-compliant CP solution.
>   - Additionally, we make several theoretical contributions for privacy-preserving online CP:
>      - Our method inherits regret guarantees from the coin-betting framework using a privacy-aware update rule $g_t \in [-1, 1]$, enabled by binary randomized response. However, achieving these guarantees is not trivial: additive-noise methods (e.g., Zhang et al., 2025) cannot ensure similar guarantees.
>      - We establish formal LDP guarantees in the fully online setting.
>      - The update rule satisfies: $\mathbb{E}[g_t] = r \cdot (\mathbb{I}\\{ S_t \le q_t \\} - (1 - \alpha)),$ which mirrors the subgradient of the pinball loss. This structure allows us to prove asymptotic coverage guarantees despite the presence of privacy noise.
>
>   - Thus, our contribution lies not only in adapting existing frameworks, but also in designing a new privacy-aware online CP algorithm for both classification and regression, with formal privacy and coverage guarantees.
>
> - **Clarification on the Comparison in Experiment Results.** We agree that DPCP (Angelopoulos et al., 2022) is an offline, central DP method and not tailored for streaming scenarios. However, we included it as a baseline to benchmark our method against existing privacy-preserving conformal predictors.
>
>    - Our experiments already span six simulation settings (with varying drift levels in both regression and classification) and two real-world datasets, providing broad evaluation under realistic online conditions.
>
>    - To address the suggestion of including a stationary setting, we conducted an additional experiment with a fixed coefficient vector ($\beta = (1, 2, 1, 0, 0)$) and no drift. Results are shown below. As expected, DPCP achieves slightly higher coverage and wider intervals. These results confirm that our method remains competitive even in stationary, offline-like settings.
>
> **Table: Proposed vs DPCP. Std values in parentheses (coverage std ×10⁻², width std ×10⁻¹)**
>
> |   | $\epsilon=3$ | $\epsilon=3$   | $\epsilon=1$ | $\epsilon=1$         | $\epsilon=0.5$  | $\epsilon=0.5$    |
> |----------|------------------------|---------------|------------------------|---------------|--------------------------|---------------|
> | Method   |  Coverage | Width | Coverage | Width  |  Coverage | Width   |
> | Proposed | 0.889 (0.3)           | 3.09 (0.4)         | 0.875 (1.0)            | 3.04 (1.0)         | 0.852 (1.9)              | 2.93 (1.8)           |
> | DPCP     | 0.904 (0.5)           | 3.33 (0.4)         | 0.911 (0.6)            | 3.40 (0.6)         | 0.922 (1.0)              | 3.53 (1.2)           |
>
>
> - **Clarification on Not Using Podkopaev et al. [2024] as Non-private Baseline**
>    - We chose DtACI as the non-private baseline because it is a well-established, widely cited method (published in JMLR, cited over 120 times), known for its stability, practicality, and reproducibility, making it a strong benchmark.
>
>    - We also carefully considered the connection to Podkopaev et al. [2024]. In fact, as explicitly noted in lines 161–164 of our paper:
>
>       > “Notably, in the special case where $r = 1$, the mechanism reduces to the non-DP setting, and the update $g_t$ degenerates to the standard subgradient $\partial \ell_{1-\alpha}(q_t, S_t)$, recovering the standard deterministic subgradient update without any privacy protection [Podkopaev et al., 2024].”
>
>    - Thus, the “non-private” curve in our experiments effectively coincides with the update structure proposed in Podkopaev et al. [2024], as it corresponds to the same subgradient-driven quantile estimation procedure under $r = 1$, i.e., without privatization. We will clarify this connection in the experimental section to avoid confusion and to highlight that our private method remains competitive even under local DP constraints.
>
> - **Comparison to Penso et al. (2025)**
>
>   - Penso et al. (2025) proposed two LDP-based CP methods: one perturbs labels using $k$-ary randomized response, and the other uses privatized conformity scores via binary indicators—both offering finite-sample coverage without accessing true labels.
>
>   - While valuable, their framework is offline, requiring a fixed calibration set and disjoint batches across multiple rounds (Algorithms 1–3). In contrast, our method is fully online, operating in a streaming setting with single-pass, $O(1)$ memory updates (Algorithm 2), making it highly efficient and scalable.
>
>   - Moreover, their approach is tailored to classification and depends on the number of labels $k$, while our method is task-agnostic, supporting both classification and regression without dependence on label cardinality.
>
> - **Comparison to Zhang et al. (2025)**
>
>   - Zhang et al. (2025) proposed an online differentially private conformal prediction (ODPCP) method using additive noise (e.g., Laplace or Gaussian) with a fixed truncation constant $c$ to ensure update stability. While effective, their approach requires careful sensitivity tuning and additional hyperparameters, complicating practical deployment.
>
>   - In contrast, our method employs randomized response to achieve LDP with only a single privacy parameter $\epsilon$, eliminating the need for extra tuning. It integrates naturally into the coin-betting framework, preserving theoretical guarantees such as bounded regret.
>
>   - Moreover, randomized response mechanisms make more efficient use of the privacy budget than additive-noise approaches, which may lead to excessive noise and degraded utility under strong privacy (Balle et al., 2018; Liu et al., 2022).
>
>   - We have conducted additional simulation studies and present a direct comparison between our method and ODPCP under various privacy levels in the following table.
>
>      - As shown, our method outperforms ODPCP under strong privacy ($\epsilon = 0.5$), offering tighter and more stable intervals. This aligns with Figure 15 in Zhang et al. (2025), which highlights ODPCP’s instability at low privacy budgets.
>
>     - In summary, we view our method as a simplification and refinement of Zhang et al. (2025), offering a more elegant, hyperparameter-free, and empirically robust framework for private online conformal prediction.
>
>    **Table: Coverages and widths for the proposed method and ODPCP**
>     *Std values in parentheses (coverage std ×10⁻², width std ×10⁻¹).*
>
> |  |   | $\epsilon=3$ | $\epsilon=3$   | $\epsilon=1$ | $\epsilon=1$         | $\epsilon=0.5$  | $\epsilon=0.5$    |
> |------|----------|------------------------|---------------|------------------------|---------------|--------------------------|---------------|
> | Case | Method   |  Coverage | Width | Coverage | Width  |  Coverage | Width   |
> | A    | Proposed | 0.889 (0.2)            | 3.25 (0.4)    | 0.875 (1.0)            | 3.16 (1.1)    | 0.853 (2.0)              | 3.11 (2.5)    |
> | A    | ODPCP    | 0.888 (0.5)            | 3.26 (0.6)    | 0.868 (1.4)            | 3.94 (10.0)   | 0.864 (2.6)              | 7.42 (54.0)   |
> | B    | Proposed | 0.889 (0.3)            | 4.16 (1.2)    | 0.874 (0.9)            | 3.94 (2.5)    | 0.851 (1.8)              | 3.60 (4.3)    |
> | B    | ODPCP    | 0.888 (0.5)            | 4.20 (1.6)    | 0.869 (1.3)            | 4.72 (11.0)   | 0.865 (2.4)              | 8.59 (54.0)   |
> | C    | Proposed | 0.889 (0.3)            | 3.10 (0.4)    | 0.875 (1.1)            | 3.04 (1.0)    | 0.852 (1.9)              | 2.95 (1.8)    |
> | C    | ODPCP    | 0.888 (0.5)            | 3.13 (0.6)    | 0.869 (1.4)            | 3.61 (23.0)   | 0.867 (2.7)              | 6.65 (30.0)   |
>
> **References**
>
> Angelopoulos et al. (2022). *Private prediction sets*. _Harvard Data Sci. Rev._, 4(2).
>
> Angelopoulos et al. (2023). *Conformal PID control for time series prediction*. _NeurIPS_, 23047–23074.
>
> Balle et al. (2018). *Privacy amplification by subsampling: Tight analyses via couplings and divergences*. _NeurIPS_.
>
> Gibbs & Candès (2021). *Adaptive conformal inference under distribution shift*. _NeurIPS_ , 1660–1672.
>
> Gibbs & Candès (2024). *Conformal inference for online prediction with arbitrary distribution shifts*. _JMLR_, 25(162):1–36.
>
> Liu et al. (2022). *Identification, amplification and measurement: A bridge to Gaussian differential privacy*. _NeurIPS_ , 11410–11422.
>
> Orabona & Pál (2016). *Coin betting and parameter-free online learning*. _NeurIPS_ .
>
> Podkopaev et al. (2024). *Adaptive conformal inference by betting*. arXiv preprint arXiv:2412.19318.

---

> > ### Comment · Reviewer_nEbK · 2025-08-04
> >
> > "We agree that DPCP (Angelopoulos et al., 2022) is an offline, central DP method and not tailored for streaming scenarios. However, we included it as a baseline to benchmark our method against existing privacy-preserving conformal predictors."
> > --> This is a point to be made in the paper. Is there any fairer state-of-art method to compare to?
> >
> > "As expected, DPCP achieves slightly higher coverage and wider intervals." + "Table: Proposed vs DPCP. Std values in parentheses (coverage std ×10⁻², width std ×10⁻¹)"
> > --> Thank you for the honest evaluation. This is a point to add in the paper that your improvement against the state of art is not uniform across all settings, but also depends on how "drifting" the distribution is where at certain point the new method is better than the old one.

---

> > > ### Author Response · Authors · 2025-08-04
> > >
> > > Thank you for your thoughtful assessment. We address each concern below.
> > >
> > > - **On Fairer Baselines and Clarification**
> > >
> > > We fully agree with your point. To avoid confusion, our original submission (lines 258–259) already stated:
> > >
> > > > "We evaluate the finite-sample performance of the proposed estimator by comparing it with two baselines: DPCP (Angelopoulos et al. [2022]), an offline private conformal method."
> > >
> > > In the revision, we will explicitly further note that DPCP serves as a benchmark for existing privacy-preserving conformal predictors, despite being offline. Following your suggestion, we reviewed prior work on privacy-preserving conformal prediction:
> > >
> > > [1] Angelopoulos et al. (2022). *Private prediction sets*. _Harvard Data Sci. Rev._
> > > -- Offline method using the exponential mechanism for DP quantile estimation; achieves finite-sample coverage under $\epsilon$-DP but requires a fixed holdout set, making it unsuitable for online or streaming settings.
> > >
> > > [2] Plassier et al. (2023). *Conformal prediction for federated uncertainty quantification under label shift*. _ICML_.
> > > -- Offline federated method (DP-FedCP) with $(\epsilon, \delta)$-DP, using importance weighting and DP-FedAvg for private quantile estimation. Requires fixed calibration data and is unsuitable for streaming scenarios.
> > >
> > > [3] Humbert et al. (2023). *One-shot federated conformal prediction*. _ICML_.
> > > -- Offline federated methods (FedCP-QQ and private FedCP²-QQ) using quantile-of-quantiles aggregation with privacy-preserving quantile correction; limited to federated settings.
> > >
> > > [4] Penso et al. (2025). *Privacy-Preserving Conformal Prediction Under Local Differential Privacy*. arXiv:2505.15721.
> > > -- Offline LDP methods: (i) label perturbation via $k$-ary randomized response, and (ii) binary indicators for privatized conformity scores. Require disjoint calibration batches and are tailored to classification, with performance dependent on label space size $k$.
> > >
> > > [5] Romanus & Molinari (2025). *Differentially Private Conformal Prediction via Quantile Binary Search*. arXiv:2507.12497.
> > > -- Offline method (P-COQS) using binary search to compute DP quantiles during calibration, ensuring privacy of prediction sets in split conformal frameworks.
> > >
> > > [6] Zhang et al. (2025). *Online Differentially Private Conformal Prediction for Uncertainty Quantification*. _ICML_.
> > > -- Online method (ODPCP) using additive noise (Laplace/Gaussian) and fixed truncation to stabilize updates; requires careful sensitivity tuning and multiple hyperparameters, complicating deployment.
> > >
> > > The following table summarizes the literature.
> > >
> > > **Table: Summary of Prior Works**
> > >
> > > | Reference | Method | Setting     | Task Type     |
> > > |-----------|-------------|-------------|---------------|
> > > | Angelopoulos et al. (2022) | Offline      | Standard    | Both          |
> > > | Plassier et al. (2023)     | Offline      | Federated   | Both    |
> > > | Humbert et al. (2023)      | Offline      | Federated   | Both    |
> > > | Penso et al. (2025)        | Offline      | Standard    | Classification |
> > > | Romanus & Molinari (2025)  | Offline      | Standard    | Both          |
> > > | Zhang et al. (2025)        | Online       | Standard    | Both          |
> > >
> > > Among these, only Zhang et al. (2025) addresses online LDP for general prediction, which we directly compare and discuss in our rebuttal.
> > >
> > > - **On Performance Variation Across Distribution Shifts**
> > >
> > > This is an excellent comment, and we fully agree. In our simulated online settings (Cases A–C), where distribution drift occurs, our method attains slightly lower coverage than DPCP but nearly halves interval width—particularly in Cases A and B—indicating strong competitive performance. This arises because DPCP relies on residuals from a fixed model trained on earlier data; under drift, residuals inflate, yielding overly conservative intervals. In a supplementary non-drifting setting, DPCP’s higher coverage is more favorable. We will make these points explicit in the revision to provide a balanced evaluation across stable and shifting environments.
> > >
> > > To better highlight the practical strengths of our method under realistic online conditions, we will also clarify the following:
> > >
> > > 1. Distribution shift in streaming settings: Offline methods like DPCP cannot adapt to evolving distributions, leading to inflated residuals and wider intervals. Our online method incrementally updates both the model and conformal threshold without storing past data, enabling real-time adaptation with greater computational efficiency.
> > >
> > > 2.  Limitations of DPCP in streaming: DPCP assumes fixed training and calibration sets, requiring full recalibration if distributions shift. This makes it incompatible with true online deployment. We include DPCP solely as a reference privacy-preserving baseline, while our method is designed for fully online, adaptive prediction with coverage guarantees and no retraining.
> > >
> > > Thank you again for your constructive comments, which have helped us produce a more focused, coherent paper.

---

### Note · Authors · 2025-08-13

We sincerely thank the reviewers, ACs, SACs, and PCs for their time, constructive feedback, and thoughtful discussions. In our rebuttal, we addressed each point in detail, and we are grateful that all reviewers expressed satisfaction with our responses—one indicated during the discussion phase the intention to raise the score from 2 to 4, another updated their score, while the rest maintained their already positive ratings. In the revised version, we will carefully incorporate the reviewers’ suggestions, including but not limited to:

- Adding comparative analyses in the experiments to provide a balanced evaluation under both stable and shifting environments;
- Reporting standard deviations for all metrics and adding a "Conclusion and Limitations" section;
- Expanding the discussion to include potential extensions to multivariate conformal prediction;
- Refining the description of Algorithm 2 to improve readability;
- Providing additional intuition in the proofs and more detailed explanations in the experiments to enhance interpretability.

We are also encouraged by the positive feedback from reviewers, summarized as follows:

- **Significance and novelty.** One reviewer described the work as "significant and original, despite the simplicity of the methods used," and another noted that "the subject is of particular interest to the CP community."
- **Theoretical investigation.** One reviewer highlighted that "the framework applies to regression and classification tasks with theoretical guarantees." Our framework provides a complete set of guarantees, including the regret bound (Corollary 3.1), privacy guarantees (Theorems 3.2–3.4), and coverage validity (Theorem 3.5).
- **Methodological soundness and clarity.** One reviewer commented that our method is "well-explained, clever, and simple in a positive manner and easy to follow," while several reviewers noted that "the paper demonstrates good writing quality."

Once again, we thank you for your insightful feedback and professional engagement throughout the review process. Guided by your valuable input, we will continue to refine and strengthen this work in future revisions.

---

### Decision · Program_Chairs · 2025-09-17

**Decision:**

Accept (poster)

**Comment:**

## Metareview

This paper studies online conformal prediction under local differential privacy (LDP). The setting is that, at each time step $t$, we receive an example $X_t$ and would like to output a set $\hat{C}_t(X_t)$ such that the label $Y_t$ is contained (i.e. "covered") in this set; we aim for a high coverage while still ensuring that the set $\hat{C}_t(X_t)$ is of small size. As usual, here we receive $Y_t$ after the prediction and we thus can update our prediction for the next rounds accordingly. In conformal prediction, the set $\hat{C}_t(X_t)$ is not arbitrary but is based on computing some score $S_t(X_t, y)$ and then keep only $y$ whose score is at most a certain threshold $q_t$. To achieve a certain coverage $1 - \alpha$, we wish to set the threshold to be the $(1 - \alpha)$-quantile. The privacy protection considered in this paper is for $(X_t, Y_t)$. The LDP model means that the algorithm immediately privatize each $(X_t, Y_t)$ and that there is no need for the curator.

The main insight in this paper is that the gradient update for $q_t$ only uses a single bit of information: whether $Y_t$ is contained in the predicted set $\hat{C}_t(X_t)$. Therefore, by applying the classic binary randomized response (and then debiasing the update to the threshold), such an algorithm can be made LDP. By subtlety combining this with online convex optimization, this gives regret guarantees for the algorithm. Finally, the authors also perform experiments; the algorithm empirically outperforms known DP conformal algorithm of Angelopoulos et al. (2022), which works in an even more relaxed model of central DP that allows a trusted curator, and is quite competitive against non-private algorithm of Gibbs and Candes (2024).

## Strengthes

- Apart from a (likely) parallel work of Zhang et al. (ICML 2025), this is the first paper that considers conformal prediction under LDP, which is a natural setting. The paper also obtains compelling initial results in the model.

- The proposed algorithm is simple and likely to be practical.

- The experimental results suggest good empirical performance, e.g. even against well-known a non-private method of Gibbs and Candes (2024).

## Weaknesses

- From LDP perspective, the use of (binary) randomized response and debiased gradient is ubiquitous in literature and is not novel, although they are still novel in the context of DP online conformal prediction.

- There are some recent papers on the topic, such as Penso et al. and Zhang et al. (ICML 2025), that are not cited or discussed in the paper. This is addressed in the rebuttal to reviewer nEbK. (Note that these papers are likely parallel contribution and might have been available online only very close to or even after the NeurIPS submission deadline.)

## Recommendation

This paper studies a new privacy-preserving model for conformal prediction and propose an algorithm that has good theoretical and empirical performance. This is thus a solid starting point that can lead to future research in this direction.